# A PRISMA systematic review of adolescent gender dysphoria literature: 3) treatment

**Lucy Thompson**[1,2,3]*, **Darko Sarovic**[1], **Philip Wilson**[3], **Louis Irwin**[3],
**Dana Visnitchi**[3], **Angela Sämfjord**[4], **Christopher Gillberg**[1,2]

**1** Gillberg Neuropsychiatry Centre (GNC), University of Gothenburg, Göteborg, Sweden, **2** Institute of Health and Wellbeing, University of Glasgow, Glasgow, United Kingdom, **3** Institute of Applied Health Science, University of Aberdeen, Inverness, United Kingdom, **4** The Child and Adolescent Psychiatric Clinic, The Queen Silvia Children's Hospital, Gothenburg, Sweden

\* Lucy.Thompson@gnc.gu.se

**Data Availability Statement:** All relevant data are within the paper.

**Funding:** The authors received no specific funding for this work.

## Abstract

It is unclear whether the literature on adolescent gender dysphoria (GD) provides evidence to inform clinical decision making adequately. In the final of a series of three papers, we sought to review published evidence systematically regarding the types of treatment being implemented among adolescents with GD, the age when different treatment types are instigated, and any outcomes measured within adolescence. Having searched PROSPERO and the Cochrane library for existing systematic reviews (and finding none at that time), we searched Ovid Medline 1946 –October week 4 2020, Embase 1947–present (updated daily), CINAHL 1983–2020, and PsycInfo 1914–2020. The final search was carried out on 2nd November 2020 using a core strategy including search terms for 'adolescence' and 'gender dysphoria' which was adapted according to the structure of each database. Papers were excluded if they did not clearly report on clinically-likely gender dysphoria, if they were focused on adult populations, if they did not include original data (epidemiological, clinical, or survey) on adolescents (aged at least 12 and under 18 years), or if they were not peer-reviewed journal publications. From 6202 potentially relevant articles (post deduplication), 19 papers from 6 countries representing between 835 and 1354 participants were included in our final sample. All studies were observational cohort studies, usually using retrospective record review (14); all were published in the previous 11 years (median 2018). There was significant overlap of study samples (accounted for in our quantitative synthesis). All papers were rated by two reviewers using the Crowe Critical Appraisal Tool v1·4 (CCAT). The CCAT quality ratings ranged from 71% to 95%, with a mean of 82%. Puberty suppression (PS) was generally induced with Gonadotropin Releasing Hormone analogues (GnRHa), and at a pooled mean age of 14.5 (±1.0) years. Cross Sex Hormone (CSH) therapy was initiated at a pooled mean of 16.2 (±1.0) years. Twenty-five participants from 2 samples were reported to have received surgical intervention (24 mastectomy, one vaginoplasty). Most changes to health parameters were inconclusive, except an observed decrease in bone density z-scores with puberty suppression, which then increased with hormone treatment. There may also be a risk for increased obesity. Some improvements were observed in global functioning and depressive symptoms once treatment was started. The most common side effects observed were acne, fatigue, changes in appetite, headaches, and mood

**Competing interests:** The authors have declared that no competing interests exist.

swings. Adolescents presenting for GD intervention were usually offered puberty suppression or cross-sex hormones, but rarely surgical intervention. Reporting centres broadly followed established international guidance regarding age of treatment and treatments used. The evidence base for the outcomes of gender dysphoria treatment in adolescents is lacking. It is impossible from the included data to draw definitive conclusions regarding the safety of treatment. There remain areas of concern, particularly changes to bone density caused by puberty suppression, which may not be fully resolved with hormone treatment.

## Introduction

This is the final of a series of three papers examining the literature on adolescent Gender Dysphoria (GD) [1, 2]. Some sections of the introductory and methodological text, and reference to methodological limitations, are necessarily repeated across all three papers. The definitions and terminology used in papers 1 and 2 were also used in the present paper [1, 2].

Gender Dysphoria (GD) is a categorical diagnosis in the Fifth Edition of the Diagnostic and Statistical Manual of Mental Disorders (DSM-5) [3]. It is also used as a general descriptive term referring to a person's discontent with assigned gender. In recent years, GD diagnoses have been increasingly made in child and adolescent services [4–6]. There has been a parallel increase in demand for gender transition interventions, particularly among natal females (NF) [4–6]. Current clinical guidance for gender transition in adolescence follows the so-called 'Dutch model', formalised in WPATH standards of care [7], where intervention is staged in accordance with a young person's age and stage of pubertal development [8, 9]. The age at which a stage of intervention will be deemed appropriate is based partly on how reversible it is. The first stage, puberty suppression (PS; prevention of the development of secondary sexual characteristics), is reversible (although not without risks to health and wellbeing) [10], the second stage, cross-sex hormone (CSH) treatment (the administration of testosterone or oestradiol to promote development of secondary sex characteristics of identified gender), is reversible to some extent (although there is a lack of evidence regarding its longer term impact) [10], and the third stage, surgical intervention, is irreversible.

Whilst it is acknowledged that any intervention during puberty has to consider the potential negative impact on young people's development, there is a surprising lack of evidence of outcomes. Research has raised safety concerns around cardiovascular health, insulin resistance, and changes to lipid profile. There have been observed cases of hypertension in NF without GD taking gonadotropin releasing hormone analogues (GnRHa) for precocious puberty [11, 12], and in three adolescent NF taking GnRHa for GD [13]. Cross-sex hormone (CSH) treatment also carries risk, with instances of an increase in thromboembolic events in adult transitioning natal males (NM) taking oral oestrogens, and worsening lipid profiles and increased insulin levels in adult transitioning NF [14]. This research has largely been conducted in adult populations, or very small samples of children / adolescents.

Given the potential risks, it is important to establish that a young person is deemed competent to make treatment decisions. This can be difficult as a key criterion is that an individual must be sufficiently mentally distressed to warrant intervention in the first place. Our previous paper demonstrated considerable mental health comorbidity among adolescents with GD [2]. The extent to which this affects treatment decision making is not clear from the existing evidence. There is evidence that most prepubertal children with GD desist once they reach puberty [15], whereas adults are more likely to persist [16]. There is some indication in the literature that adolescents are unlikely to desist [17, 18] and recent data from Butler et al. (2022)

showed in a UK sample that desistance was more likely in those aged 15 years or under than those 16 or over (9.2% vs 4.4%) [19]. But there is a lack of relevant recent follow-up studies. There is an explicit lack of evidence on adolescent-onset GD, so the interplay between GD and other mental health (MH) factors in this phenomenon is not well understood [20].

Intense international debate regarding a number of issues relating to GD intervention in adolescence is ongoing, especially within Europe and North America where the main research active treatment centres are based [20]. One recent high profile legal case (Bell vs Tavistock [21]) attracted considerable attention from people and organisations with a range of strongly-held views both in favour of and against the ruling, illustrating the acknowledged lack of good quality evidence regarding treatment comorbidities and outcomes to inform service design [22, 23]. Concern in the UK led to the commissioning of the Cass review, which in March 2022 recommended the Tavistock Gender and Identity Development Service (GIDS) be closed down in favour of developing regional specialist centres [24] and highlighted the need not only for much better evidence but also for clinicians to be better informed and willing to work toward positive change [25]. Services are sometimes left having to make unilateral decisions against national guidance, e.g., the Karolinska University Hospital in Stockholm, Sweden, changing their policy to limit puberty suppression to the context of clinical studies [26], and recently a new version of World Professional Association for Transgender Health (WPATH) standards of care have removed the lower age limit for certain treatments, leaving this decision to clinician judgement [27, 28].

## Scope of the review

This review is the third in a three-part series addressing the current state of evidence on gender dysphoria experienced in adolescence. Our over-arching aim was to establish what the literature tells us about gender dysphoria in adolescence. We broke this down into seven specific questions (see below). Paper 1 [1] addressed questions 1-3c (*italicised*), Paper 2 [2] addressed question 4 (plain text), and Paper 3, the current paper, addresses questions 3d and 5–7 (**bold text**). Our overall research aim was addressed by seven specific questions:

1. *What is the prevalence of GD in adolescence*?

2. *What are the proportions of natal males / females with GD in adolescence (a) and has this changed over time (b)*?

3. **What is the pattern of age at** *(a) onset (b) referral (c) assessment* **(d) treatment?**

4. What is the pattern of mental health problems in this population?

5. **What treatments have been used to address GD in adolescence?**

6. **What outcomes are associated with treatment/s for GD in adolescence?**

7. **What are the long-term outcomes for all (treated or otherwise) in this population?**

The present paper focuses on questions 3d, 5–7. We have addressed questions 1, 2, 3a, 3b, and 3c in our first paper [1], and question 4 in paper 2 [2]. The methodology below includes the searches conducted for the whole review.

We set out to include any paper offering primary data in response to any of these questions.

## Methods

### Protocol and registration

The systematic review protocol was submitted to PROSPERO on the 28th November 2019, and registered on 17 March 2020 (registration number CRD42020162047). An update was

uploaded on 2[nd] February 2021 to include specific detail on age criteria and clinical verification of condition. The review has been prepared according to PRISMA 2020 [29] guidelines (see Table 1 for checklist).

## Eligibility criteria

The volume of non-peer-reviewed literature in initial searches proved so great that we took the decision to only include peer-reviewed journal papers featuring original research data. This decision was made subsequent to initial PROSPERO registration, but prior to full text screening. Complete inclusion criteria were:

- Focused on gender dysphoria or transgenderism;

- Includes data on adolescents (aged 12–17 years inclusive);

- Includes original data (not review paper or opinion piece);

- Peer-reviewed publication (not theses or conference proceedings);

- In English language.

## Information sources

We searched PROSPERO and the Cochrane library for existing systematic reviews. We searched Ovid Medline 1946 –October week 4 2020, Embase 1947–present (updated daily), CINAHL 1983–2020, and PsycInfo 1914–2020. After selecting the final sample of articles, the first author used their reference lists as a secondary data source.

## Search

The final search was carried out on 2[nd] November 2020 using a core strategy which was adapted according to the structure of each database. The core strategy included search terms for 'adolescence' and 'gender dysphoria'. This was kept deliberately broad in order to ensure any studies on the subject could be screened for eligibility. The specific search strategy employed in EMBASE is given below, and represents the format followed with the others. The specific search strategies employed in each database are detailed in Table 2.

## EMBASE search

1. Exp adolescence/

2. (adolesc* or teen* or puberty*).tw.

3. 1 or 2

4. exp gender dysphoria/

5. exp transgender/

6. sex reassignment/

7. (gender dysphoria or gender identity or transsex* or trans sex or transgender or trans gender or sex reassignment).tw.

8. 4 or 5 or 6 or 7

9. 3 and 8

**Table 1.** PRISMA 2020 checklist.

| Section and Topic | Item # | Checklist item | Location where item is reported |
|---|---|---|---|
| **TITLE** | | | |
| Title | 1 | Identify the report as a systematic review. | 1 |
| **ABSTRACT** | | | |
| Abstract | 2 | See the PRISMA 2020 for Abstracts checklist. | 2 |
| **INTRODUCTION** | | | |
| Rationale | 3 | Describe the rationale for the review in the context of existing knowledge. | 4 |
| Objectives | 4 | Provide an explicit statement of the objective(s) or question(s) the review addresses. | 5 |
| **METHODS** | | | |
| Eligibility criteria | 5 | Specify the inclusion and exclusion criteria for the review and how studies were grouped for the syntheses. | 6–7 |
| Information sources | 6 | Specify all databases, registers, websites, organisations, reference lists and other sources searched or consulted to identify studies. Specify the date when each source was last searched or consulted. | 6 |
| Search strategy | 7 | Present the full search strategies for all databases, registers and websites, including any filters and limits used. | 6, Table 2 |
| Selection process | 8 | Specify the methods used to decide whether a study met the inclusion criteria of the review, including how many reviewers screened each record and each report retrieved, whether they worked independently, and if applicable, details of automation tools used in the process. | 7–8 |
| Data collection process | 9 | Specify the methods used to collect data from reports, including how many reviewers collected data from each report, whether they worked independently, any processes for obtaining or confirming data from study investigators, and if applicable, details of automation tools used in the process. | 8 |
| Data items | 10a | List and define all outcomes for which data were sought. Specify whether all results that were compatible with each outcome domain in each study were sought (e.g. for all measures, time points, analyses), and if not, the methods used to decide which results to collect. | 8 |
| | 10b | List and define all other variables for which data were sought (e.g. participant and intervention characteristics, funding sources). Describe any assumptions made about any missing or unclear information. | 8 |
| Study risk of bias assessment | 11 | Specify the methods used to assess risk of bias in the included studies, including details of the tool(s) used, how many reviewers assessed each study and whether they worked independently, and if applicable, details of automation tools used in the process. | 8 |
| Effect measures | 12 | Specify for each outcome the effect measure(s) (e.g. risk ratio, mean difference) used in the synthesis or presentation of results. | 8 |
| Synthesis methods | 13a | Describe the processes used to decide which studies were eligible for each synthesis (e.g. tabulating the study intervention characteristics and comparing against the planned groups for each synthesis (item #5)). | 8 |
| | 13b | Describe any methods required to prepare the data for presentation or synthesis, such as handling of missing summary statistics, or data conversions. | 9 |
| | 13c | Describe any methods used to tabulate or visually display results of individual studies and syntheses. | 8–9 |
| | 13d | Describe any methods used to synthesize results and provide a rationale for the choice(s). If meta-analysis was performed, describe the model(s), method(s) to identify the presence and extent of statistical heterogeneity, and software package(s) used. | 8–9 |
| | 13e | Describe any methods used to explore possible causes of heterogeneity among study results (e.g. subgroup analysis, meta-regression). | N/A |
| | 13f | Describe any sensitivity analyses conducted to assess robustness of the synthesized results. | N/A |
| Reporting bias assessment | 14 | Describe any methods used to assess risk of bias due to missing results in a synthesis (arising from reporting biases). | N/A |
| Certainty assessment | 15 | Describe any methods used to assess certainty (or confidence) in the body of evidence for an outcome. | N/A |
| **RESULTS** | | | |
| Study selection | 16a | Describe the results of the search and selection process, from the number of records identified in the search to the number of studies included in the review, ideally using a flow diagram. | 7, Fig 1 |
| | 16b | Cite studies that might appear to meet the inclusion criteria, but which were excluded, and explain why they were excluded. | Table 4 |

*(Continued)*

**Table 1.** (Continued)

| Section and Topic | Item # | Checklist item | Location where item is reported |
|---|---|---|---|
| Study characteristics | 17 | Cite each included study and present its characteristics. | Table 3 |
| Risk of bias in studies | 18 | Present assessments of risk of bias for each included study. | Table 9; Fig 2 |
| Results of individual studies | 19 | For all outcomes, present, for each study: (a) summary statistics for each group (where appropriate) and (b) an effect estimate and its precision (e.g. confidence/credible interval), ideally using structured tables or plots. | Tables 5–8 |
| Results of syntheses | 20a | For each synthesis, briefly summarise the characteristics and risk of bias among contributing studies. | 9–10 |
| | 20b | Present results of all statistical syntheses conducted. If meta-analysis was done, present for each the summary estimate and its precision (e.g. confidence/credible interval) and measures of statistical heterogeneity. If comparing groups, describe the direction of the effect. | N/A |
| | 20c | Present results of all investigations of possible causes of heterogeneity among study results. | N/A |
| | 20d | Present results of all sensitivity analyses conducted to assess the robustness of the synthesized results. | N/A |
| Reporting biases | 21 | Present assessments of risk of bias due to missing results (arising from reporting biases) for each synthesis assessed. | N/A |
| Certainty of evidence | 22 | Present assessments of certainty (or confidence) in the body of evidence for each outcome assessed. | N/A |
| **DISCUSSION** | | | |
| Discussion | 23a | Provide a general interpretation of the results in the context of other evidence. | 16 |
| | 23b | Discuss any limitations of the evidence included in the review. | 17–18 |
| | 23c | Discuss any limitations of the review processes used. | 17–18 |
| | 23d | Discuss implications of the results for practice, policy, and future research. | 18 |
| **OTHER INFORMATION** | | | |
| Registration and protocol | 24a | Provide registration information for the review, including register name and registration number, or state that the review was not registered. | 6 |
| | 24b | Indicate where the review protocol can be accessed, or state that a protocol was not prepared. | 6 |
| | 24c | Describe and explain any amendments to information provided at registration or in the protocol. | N/A |
| Support | 25 | Describe sources of financial or non-financial support for the review, and the role of the funders or sponsors in the review. | Submission system |
| Competing interests | 26 | Declare any competing interests of review authors. | 19 |
| Availability of data, code and other materials | 27 | Report which of the following are publicly available and where they can be found: template data collection forms; data extracted from included studies; data used for all analyses; analytic code; any other materials used in the review. | Template: 8 Data: Tables 5–8 |

*From*: Page MJ, McKenzie JE, Bossuyt PM, Boutron I, Hoffmann TC, Mulrow CD, et al. The PRISMA 2020 statement: an updated guideline for reporting systematic reviews. BMJ 2021;372:n71. doi: 10.1136/bmj.n71

## Study selection

The study selection process is illustrated in Fig 1. We used Endnote v. X7.8 to manage all references, and followed the de-duplication and management strategies set out in Bramer et al. (2016) [30] and Peters (2017) [31] respectively.

In the first stage of screening, papers were excluded based on their title or abstract if they did not clearly report on gender dysphoria or transgenderism and if they were focused on adult populations. In the second stage of screening, papers were excluded on the basis of title and abstract if they did not include original data (epidemiological, clinical, or survey) on adolescents (aged at least 12 and under 18 years). At both stages papers were retained if there was insufficient information to exclude them.

Full-text files were obtained for the remaining records.

Papers were rejected at this stage if they:

**Table 2. Search terms.**

| | EMBASE | Ovid Medline | CINAHL | PsycInfo |
|---|---|---|---|---|
| **Adolescence** | 1. Exp adolescence/<br>2. (adolesc* or teen* or puberty*).tw. | 1. Exp adolescence/<br>2. (adolesc* or teen* or puberty*).tw. | 1. (MH "Adolescence+")<br>2. TI adolesc* OR TI teen* OR TI pubert* OR AB adolesc* OR AB teen* OR AB pubert* | 1. TI adolescence OR AB adolescence<br>2. TI adolesc* OR TI teen* OR TI pubert*<br>3. AB adolesc* OR AB teen* OR AB pubert* |
| **Gender Dysphoria** | 3. exp gender dysphoria/<br>4. exp transgender/<br>5. sex reassignment/<br>6. (gender dysphoria or gender identity or transsex* or trans sex or transgender or trans gender or sex reassignment).tw. | 2. exp gender dysphoria/<br>3. exp transgender/<br>4. Exp Sex Reassignment Procedures/<br>5. (gender dysphoria or gender identity or transsex* or trans sex or transgender or trans gender or sex reassignment).tw. | 3. (MH "Gender Dysphoria")<br>4. (MH "Transgender Persons") OR (MH "Transsexuals")<br>5. (MH "Sex Reassignment Procedures+")<br>6. TI gender dysphoria OR AB gender dysphoria OR TI gender identity disorder OR AB gender identity disorder OR TI transsex* OR AB transsex* OR TI trans sex* OR AB trans sex* OR TI transgender OR AB transgender OR TI trans gender OR AB trans gender<br>7. TI sex reassignment OR AB sex reassignment OR TI gender reassignment OR AB gender reassignment | 4. DE "Gender Dysphoria" OR DE "Gender Nonconforming" OR DE "Gender Reassignment" OR DE "Gender Identity" OR DE "Transsexualism" OR DE "Transgender"<br>5. TI gender dysphoria OR TI gender identity disorder OR TI transsex* OR TI trans sex* OR TI transgender OR TI trans gender OR TI sex reassignment OR TI gender reassignment<br>6. AB gender dysphoria OR AB gender identity disorder OR AB transsex* OR AB trans sex* OR AB transgender OR AB trans gender OR AB sex reassignment OR AB gender reassignment |
| **Combination terms** | 7. 1 OR 2<br>8. 3 OR 4 OR 5 OR 6<br>9. 7 AND 8 | 7. 1 OR 2<br>8. 3 OR 4 OR 5 OR 6<br>9. 7 AND 8 | 8. 1 OR 2<br>9. 3 OR 4 OR 5 OR 6 OR 7<br>10. 8 AND 9 | 7. 1 OR 2 OR 3<br>8. 4 OR 5 OR 6<br>9. 7 AND 8 |

- Contained no original data (including literature and clinical reviews, journalistic / editorial pieces, letters and commentaries);

- Included only case studies or selected case series;

- Pertained to conditions other than GD (e.g., Disorders of sexual development or HIV);

- Did not include clinically-identified GD (e.g., survey where participants self-identify, with no clinical contact);

- Pertained to populations other than those with GD (e.g., LGBTQ more broadly);

- Pertained to populations including or restricted to those aged 18 years or older. This included papers where adolescents and adults were included in the same sample, but adolescents were not separately reported (in many cases age range was not reported and so a 'balance of probabilities' assessment had to be made based on the reported mean age);

- Pertained to populations restricted to those aged under 12 years of age. This included papers where adolescents and children were included in the same sample, but the majority of participants were clearly under 12 (based on mean or median age);

- Where participants were practitioners, not patients;

- Referred only to conference proceedings;

- Were written in a non-European language (e.g., Turkish);

- Could not be obtained (including due to being published in non-English language journals, or in theses).

**Table 3. Study characteristics.**

| ID | Country | Reference | Design | Setting | N | Age (years) | Male natal sex (%) | Date range | GD status | Treatment stage 1 | 2 | 3 |
|----|---------|-----------|--------|---------|---|-------------|--------------------|-----------|-----------|-------------------|---|---|
| 1 | Belgium | Tack, et al. (2016). Consecutive lynestrenol and cross-sex hormone treatment in biological female adolescents with gender dysphoria: A retrospective analysis. Hormone Research in Paediatrics, 86 (Supplement 1), 268–269. | obs, retro, longit, interv | Division of Pediatric Endocrinology, Ghent University*, Belgium | 43 | mean age start of progestins 15.8 mean age start of CSH 17.4 | 0 | 2010–2015 | 2 | ▓ | ▓ | |
| 2 | | Tack, et al. (2017). Consecutive Cyproterone Acetate and Estradiol Treatment in Late-Pubertal Transgender Female Adolescents. Journal of Sexual Medicine, 14(5), 747–757. | obs, retro, longit, interv | Division of Pediatric Endocrinology, Ghent University*, Belgium | 27 | Mean age at start of CA 16y6m, CA +oestradiol 17y7m | 100.0 | 2008–2016 | 1 | ▓ | ▓ | |
| 3 | | Tack, et al. (2018). Proandrogenic and Antiandrogenic Progestins in Transgender Youth: Differential Effects on Body Composition and Bone Metabolism. Journal of Clinical Endocrinology and Metabolism, 103(6), 2147–2156. | obs, retro, longit, interv | Division of Pediatric Endocrinology, Ghent University* | 65[$] | Mean NF: $16·2\pm1·05$ Mean NM:$16·3$ $\pm1·21$ | 32·3 | 2011–2017 | 1 | ▓ | | |
| 4 | Germany | Becker-Hebly, et al. (2020). Psychosocial health in adolescents and young adults with gender dysphoria before and after gender-affirming medical interventions: a descriptive study from the Hamburg Gender Identity Service. European Child and Adolescent Psychiatry. | obs, prosp, longit, interv | Gender Identity Service, University Medical Center Hamburg-Eppendorf, Germany | 54 | 11.21–17.34 | 14.8 | Sept 2013 – Jun 2017 | 1 | ▓ | ▓ | ▓ |
| 5 | Israel | Perl, et al. (2020). Blood Pressure Dynamics After Pubertal Suppression with Gonadotropin-Releasing Hormone Analogs Followed by Testosterone Treatment in Transgender Male Adolescents: A Pilot Study. Lgbt Health, 7(6), 340–344. | obs, retro, longit | Gender Dysphoria Clinic at Dana-Dwek Children's Hospital, Gender Clinic, Tel Aviv Sourasky Medical Center, Israel | 15 | GnRHa group (n15): $14.4\pm1.0$ GAH group (n9): $15.1\pm0.9$ | 100 | 2013–2018 | 2 | ▓ | ▓ | ▓ |

*(Continued)*

**Table 3.** (Continued)

| ID | Country | Reference | Design | Setting | N | Age (years) | Male natal sex (%) | Date range | GD status | Treatment stage | | |
|----|---------|-----------|--------|---------|---|-------------|-------------------|------------|-----------|---|---|---|
| | | | | | | | | | | 1 | 2 | 3 |
| 6 | Nether-lands | de Vries, et al. (2011). Puberty suppression in adolescents with gender identity disorder: A prospective follow-up study. Journal of Sexual Medicine, 8(8), 2276–2283. | obs & comp, prosp, longit | VU University Medical Center, Amsterdam (forerunner to CEGD) | 70 | 11·1–17·0 | 47·1 | 2000–2008 | 1 | | | |
| 7 | Nether-lands | Klaver, et al. (2018). Early Hormonal Treatment Affects Body Composition and Body Shape in Young Transgender Adolescents. Journal of Sexual Medicine, 15(2), 251–260. | obs, retro, longit, interv | VU University Medical Center, Amsterdam (forerunner to CEGD) | 192 | Mean at start puberty blocker NM 14.5±1.8 NF 15.3±2.0 | 37 | 1998–2014 | 1 | | | |
| 8 | Nether-lands | Klaver, et al. (2020). Hormonal Treatment and Cardiovascular Risk Profile in Transgender Adolescents. Pediatrics, 145(3), 03. | obs, retro, longit, interv | VU University Medical Center, Amsterdam (forerunner to CEGD) | 192 | Mean at start of puberty blocker NM 14.6±1.8 NF 15.2±2.0 | 37 | 1998–2015 | 1 | | | |
| 9 | Nether-lands | Schagen, et al. (2020). Bone Development in Transgender Adolescents Treated With GnRH Analogues and Subsequent Gender-Affirming Hormones. Journal of Clinical Endocrinology & Metabolism, 105(12), 01. | obs, prosp, longit, interv | VU University Medical Center, Amsterdam (forerunner to CEGD)*, Netherlands | 121 | NM 14.1±1.7 NF 14.5±2.0 | 42 | Eligible for treatment 1998–2009 | 1 | | | |
| 10 | Nether-lands | Stoffers, I. E., de Vries, M. C., & Hannema, S. E. (2019). Physical changes, laboratory parameters, and bone mineral density during testosterone treatment in adolescents with gender dysphoria. Journal of Sexual Medicine, 16(9), 1459–1468. | obs, retro, x-sect, interv | not stated, but assume Center of Expertise on Gender Dysphoria (CEGD), Amsterdam | 62 NF | 11.8–18.0 | N/A | Nov 2010-Aug 2018 | 1 | | | |
| 11 | UK | Costa, et al. (2015). Psychological Support, Puberty Suppression, and Psychosocial Functioning in Adolescents with Gender Dysphoria. Journal of Sexual Medicine, 12(11), 2206–2214. | obs & comp, retro, longit, interv | Gender Identity Development Service, Tavistock & Portman, London | 201 (Control: 169 CAMHS Stockholm cases) | 12–17 | 37·8 | 2010–2014 | 1 | | | |

(*Continued*)

**Table 3.** (Continued）

| ID | Country | Reference | Design | Setting | N | Age (years) | Male natal sex (%) | Date range | GD status | Treatment stage | | |
|----|---------|-----------|--------|---------|---|-------------|--------------------|------------|-----------|---|---|---|
| | | | | | | | | | | 1 | 2 | 3 |
| 12 | UK | Joseph, et al. (2019). The effect of GnRH analogue treatment on bone mineral density in young adolescents with gender dysphoria: findings from a large national cohort. Journal of pediatric endocrinology & metabolism: JPEM., 31. | obs, retro, longit, interv | Gender Identity Development Service, Tavistock & UCLH Early Intervention programme @ national endocrine clinic, London | 70 | 12–14 | 44·3 | 2011–2016 | 2 | ▨ | | |
| 13 | UK | Russell, et al. (2020). A Longitudinal Study of Features Associated with Autism Spectrum in Clinic Referred, Gender Diverse Adolescents Accessing Puberty Suppression Treatment. Journal of Autism and Developmental Disorders. | obs, retro, x-sect | Gender Identity Service, Tavistock & Portman, London | 95 | 9·9–15·9 (mean 13·6 ±0·11) | 40 | Not given | 2 | ▨ | | |
| 14 | USA | Chen, et al. (2016). Characteristics of Referrals for Gender Dysphoria over a 13-Year Period. Journal of Adolescent Health, 58 (3), 369–371. | obs, retro, x-sect | paediatric endocrinology clinic, Indiana | 38 | Mean age 14·4 ±3·2 | 42·1 | 1/1/02–1/4/15 | 3 | | ▨ | |
| 15 | USA | Jensen, et al. (2019). Effect of Concurrent Gonadotropin-Releasing Hormone Agonist Treatment on Dose and Side Effects of Gender-Affirming Hormone Therapy in Adolescent Transgender Patients. Transgender Health, 4 (1), 300–303. | obs, retro, x-sect | 'a pediatric gender clinic at a tertiary medical center', Ann & Robert H. Lurie Children's Hospital of Chicago* | 17 | 11.4–15.6 | 35.3 | Treatment before Mar 2016, data extracted to end Jan 2018 | 2 | ▨ | | |
| 16 | USA | Kuper, et al. (2020). Body Dissatisfaction and Mental Health Outcomes of Youth on Gender-Affirming Hormone Therapy. Pediatrics, 145(4), 04 | obs, prosp, longit, interv | 'a multidisciplinary program in Dallas, Texas' | 148 | 9–18 | 37 | Initially assessed 2014–2018 | 1 | ▨ | | ▨ |
| 17 | USA | Lee, et al. (2020). Low Bone Mineral Density in Early Pubertal Transgender/Gender Diverse Youth: Findings From the Trans Youth Care Study. Journal of the Endocrine Society, 4 (9), 1–12. doi:10.1210/jendso/bvaa065 | obs, prosp, x-sect | Trans Youth Care Study: Children's Hospital Los Angeles, Lurie Children's Hospital, Boston Children's Hospital, and University of California San Francisco Benioff Children's Hospital | 63 | Mean at start of puberty blocker NM 12.1±1.3 NF 11.0±1.4 | 52.4 | Not given | 1* | ▨ | | |

*(Continued)*

**Table 3.** (Continued)

| ID | Country | Reference | Design | Setting | N | Age (years) | Male natal sex (%) | Date range | GD status | Treatment stage | | |
|----|---------|-----------|--------|---------|---|-------------|---------------------|------------|-----------|---|---|---|
| | | | | | | | | | | 1 | 2 | 3 |
| 18 | USA | Lopez, et al. (2018). Trends in the use of puberty blockers among transgender children in the United States. Journal of Pediatric Endocrinology and Metabolism, 31(6), 665–670. | obs, retro, x-sect | US Pediatric Health and Information System (PHIS) database | 40[a] | 8·8–17·8[a] | 46·3 | 2010-2015[a] | 3 | ▓ | | |
| 19 | USA | Nahata, et al. (2017). Mental Health Concerns and Insurance Denials Among Transgender Adolescents. LGBT Health, 4(3), 188–193. | obs, retro, x-sect | 'gender program' at an 'urban, Mid-western, pediatric academic center', Nationwide Children's Hospital, Columbus, Ohio* | 79 | 9–18 | 35·4 | 2014–2016 | 3 | ▓ | | |

Key

* = author's affiliation. No specific setting given  obs = observational  comp = comparative

$ = authors report sample overlap  prosp = prospective  retro = retrospective

a = Max age of whole sample 18.8. Only data on those under 18 years used in this review. longit = longitudinal  x-sect = cross-sectional

interv = intervention study

GD status codes: 1) clinical diagnosis using DSM-III / IV / IV-TR / 5; 2) active patients within clinic; 3) data mined using ICD 9 / 10 codes and/or relevant keywords; 4) Under assessment at clinic–beyond referral stage.

Treatment stage: 1 = puberty suppression; 2 = cross sex hormones; 3 = surgical intervention

Following initial full text screening, all remaining papers were assessed by a second reviewer to reduce the risk of inclusion bias. Where reviewers reached a different conclusion, discussion took place to reach consensus. If agreement could not be reached, a third reviewer was consulted, and discussion used to reach consensus amongst all three reviewers.

Data extracted from eligible papers were tabulated and used in the qualitative synthesis. Given the limited number of specialist treatment centres globally, we assessed how many of the included papers featured the same or overlapping samples.

Papers included in the sub-sample for the present analysis contained some data on either age at treatment commencement, type of treatment administered, or treatment outcomes.

## Quality assessment

All papers were rated by two reviewers using the Crowe Critical Appraisal Tool v1·4 (CCAT [32]). CCAT is suitable for a range of methodological approaches, assessing papers in terms of eight categories: Preliminaries (overall clarity and quality); Introduction; Design; Sampling; Data collection; Ethical matters; Results; Discussion. Each category is rated out of 5 and all eight categories summed to give a total out of 40 (converted to a percentage). In the present review, each paper was then assigned to one of five categories, based on the average rating of the reviewers, where a rating of 0–20% was coded 1 (poorest quality), and 81–100% coded 5 (highest quality). Inter-rater reliability was shown to be very good ($k = 0.93$, SE = 0·07).

## Data collection process

Data were extracted from the papers using the CCAT form (https://conchra.com.au/wp-content/uploads/2015/12/CCAT-form-v1.4.pdf) by two reviewers per paper and compiled by

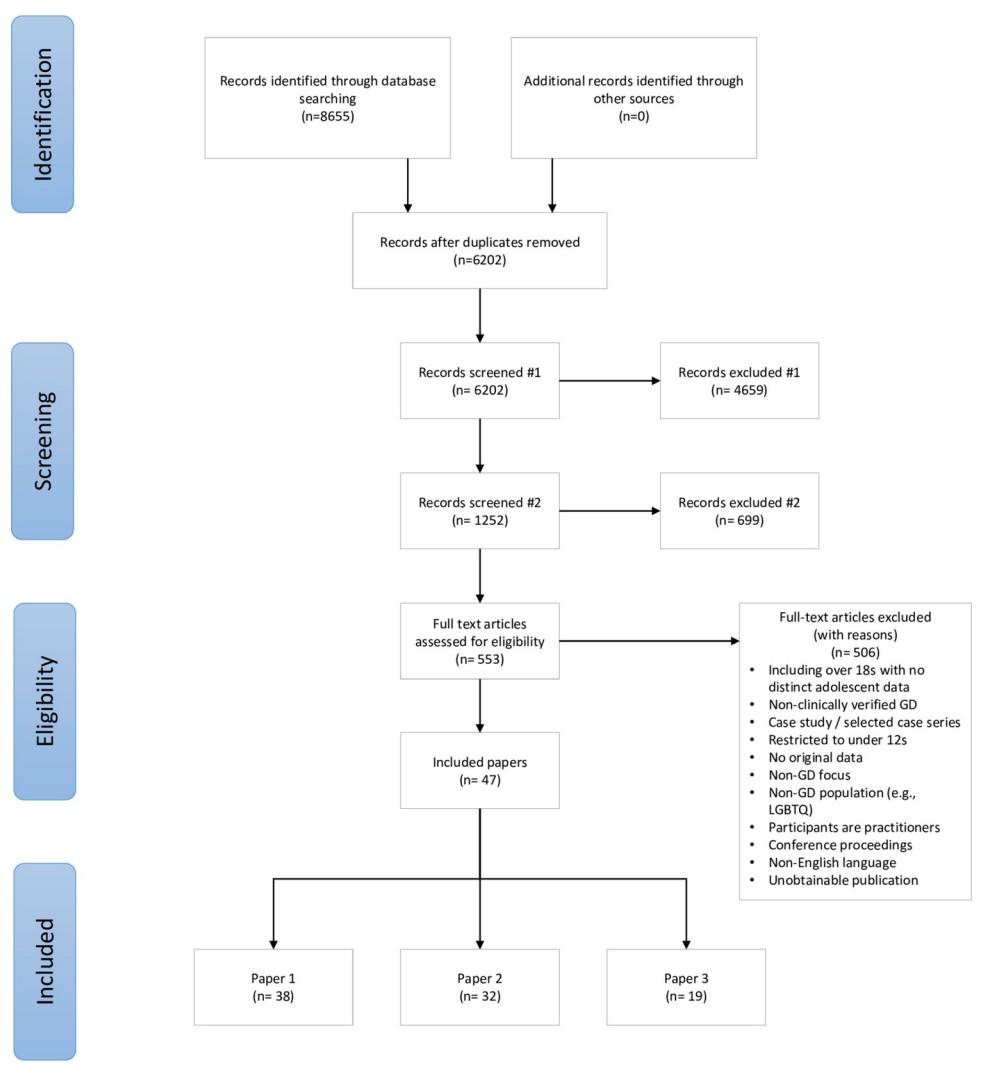

**Fig 1. PRISMA diagram.**

the first author (LT). Once compiled, instances of overlap between papers (i.e., if the same sample was described in two papers) were identified and tabulated, and the final sample for each question defined.

## Results

### Number of studies included, retained and excluded

The PRISMA diagram in Fig 1 provides details of the screening and exclusion process. The searches returned 8655 results, reduced to 6202 following de-duplication. Titles and abstracts were screened by one reviewer (LT) and 4659 records excluded after initial screening and a further 699 excluded on second stage title / abstract screening. This left 553 eligible for full text screening. An initial screening (LT) of full texts reduced the number of records to 155. Forty-seven papers were included in the final dataset, of which 19 included data for the present paper. Full characteristics of included studies are provided in Table 3.

## Study characteristics

All of the included studies originated from a small number of centres in wealthy nations: USA (n = 6), Netherlands (n = 5), UK (n = 3), Belgium (n = 3), Germany (n = 1), and Israel (n = 1). The Belgian studies came from the same centre, an adolescent gender clinic in Ghent. The Dutch studies were from the same centre in Amsterdam. The UK studies consisted of samples assessed at the Gender Identity Development Service in London. As such there may be overlap in the samples studied but this was not always clearly reported. Both Klaver et al. papers [33, 34] examine the same cohort but investigate different parameters. There is partial overlap reported in Tack. et al. (2018) [35] with the previous two Tack et al. (2016 and 2017) papers [36, 37], however the degree of overlap is not fully described. Overlap was therefore estimated using dates of record search or dates of inclusion. Fig 2 provides a graphic representation of likely sample overlap.

All papers included were published within the last eleven years, with the earliest having been published in 2011 (de Vries et al. [38]). Reported dates of treatment ranged from 1998 to 2018 inclusive. The majority (n = 13) of studies were retrospective in nature, and predominantly took their data from review of medical records. None of the studies included in this paper were randomised controlled trials (RCTs).

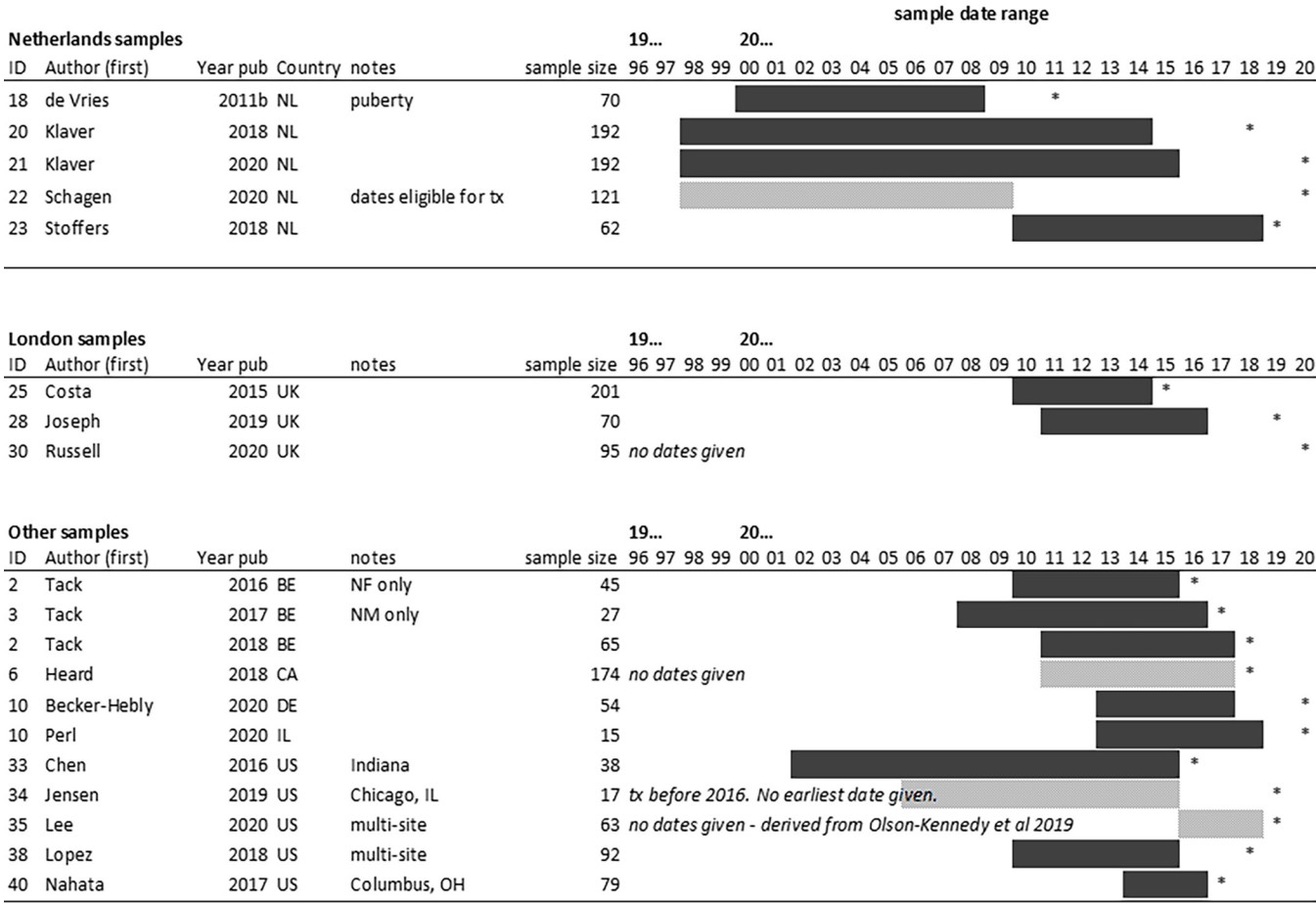

**Fig 2. Overlap between included samples.** Key: tx: treatment. *: year of publication. NL: Netherlands; UK: United Kingdom; BE: Belgium; CA: Canada; DE: Germany; IL: Israel; US: United States.

Most papers (n = 14) contained both NF and NM participants. Three (Perl et al. [39], Stoffers et al. [40], Tack et al., 2016 [36]) contained data pertaining to NF only, and one (Tack et al., 2017 [37]) contained data pertaining to NM only. In total, approximately 1300 adolescents who had been treated for GD were included in this analysis. Of this, around 1102 were treated with puberty suppression (PS), and 727 were treated with CSH, either in addition to PS (n = 506) or as monotherapy (n = 221).

All nineteen papers included data on the treatments offered for adolescent GD. All nineteen included data on PS, of which six [35, 41–45] focused on this exclusively. Of those focusing exclusively on PS, four [42–45] analysed GnRHa use, one (Tack et al. 2018) [35] analysed the effects of progestins exclusively, and one (Lee et al.) only contained treatment used and age of treatment data [41]. Thirteen papers looked at the effects of both PS and CSH. Of these, nine looked at the effects of GnRHa and CSH, two looked at the effects of progestins and CSH [36, 37] and one (Chen et al.) [46] looked at the effects of GnRHa, progestins, androgen-receptor blockers and CSH. Eight analysed both oestrogen and testosterone therapies, and two [39, 40] analysed testosterone solely.

Patients in most papers (n = 15) had a GD diagnosis according to either the DSM-IV or V definition of Gender Dysphoria. Two papers [42, 47] used ICD-9/10 definitions. Perl et al. [39] did not describe how patients were diagnosed with GD, but did state the participants had sought out and had been treated for GD at a Paediatric Gender Dysphoria Clinic. Likewise Jensen et al. [48] used data from adolescent participants who had received or were receiving CSH therapy at a paediatric gender clinic for GD.

A substantial group of papers narrowly missed inclusion criteria, mostly on the age criterion and some on the clinically likely GD criterion, and were not included in the final sample of reviewed papers. We documented characteristics of all studies excluded at the final full text screen in Table 4.

## Overall findings based on included studies

**What is the pattern of age at treatment? What treatments have been used to address GD in adolescence?.** The age of initiation of PS was explicitly stated in 17 papers. After accounting for sample overlap and only including those papers reporting mean and SD, the final age calculation is from 11 papers. The pooled mean age at which PS was started was 14.5 (±1.0) years, with the lowest treatment age 8.8 years old [42]. Only five papers were included in the calculation of mean age of initiation of CSH therapy (16.2±1.0). The lowest age of CSH initiation was reported at 13.2 years [49]. This mean included those who initially had PS monotherapy before adding CSH therapy, and those who were started on CSH monotherapy. Age data used in this analysis are presented in Tables 5 and 6.

The method of PS was fairly consistent for this age group. Most (15/19) papers described the use of GnRHa. Only three (all Tack et al. [35–37]) did not use GnRHa, however they did state that common practice in their centre is to prescribe GnRHa to patients who present at a less advanced stage of puberty. Kuper et al. [49] does not record what method was used for PS. In Chen et al. (2016), only 39.5% (n = 15) received PS treatment. Of those 15, 6 (40%) were denied insurance approval for GnRHa, so second-line alternatives, i.e., progestins and anti-androgens, were prescribed. From the patients who received PS treatment, only 21% were prescribed CSH at a recommended age (around 16 years old). Most studies with CSH data included both NF and NM participants, so the effects of both oestradiol and testosterone could be reported. One study indicated that, as expected, lower doses of oestradiol and testosterone cypionate were required for individuals who had GnRHa before starting with the CSH treatment [48].

**Table 4. Papers excluded at second full text screen (i.e., closely missed meeting inclusion criteria).**

| ID | Reference | Date | Location & setting | Reason for exclusion | Notes |
|----|-----------|------|--------------------|-----------------------|-------|
| 1 | Achille, C., Taggart, T., Eaton, N. R., Osipoff, J., Tafuri, K., Lane, A., & Wilson, T. A. (2020). Longitudinal impact of gender-affirming endocrine intervention on the mental health and well-being of transgender youths: preliminary results. International Journal of Pediatric Endocrinology, 2020, 8. | 2020 | USA<br>New York, Stoney Brook Children's Hospital | 1<br>Mean age 16.2±2.2 | 66% NF<br>MH improved with endocrine intervention<br>Small sample (n = 50) |
| 2 | Aitken, et al. (2015). Evidence for an altered sex ratio in clinic-referred adolescents with gender dysphoria. Journal of Sexual Medicine, 12(3), 756–763. | 2015 | 1) CANADA<br>Gender Identity Service, Child, Youth and Family Services (CYFS), Toronto<br>2) NETHERLANDS<br>Center of Expertise on Gender Dysphoria (CEGD) | 1, 2<br>1) unclear if GD clinically verified<br>2) Eldest participant 19 years | Useful information in change of gender in those presenting to services over time. |
| 3 | Akgul, G. Y., Ayaz, A. B., Yildirim, B., & Fis, N. P. (2018). Autistic Traits and Executive Functions in Children and Adolescents With Gender Dysphoria. Journal of sex & marital therapy, 44(7), 619–626. | 2018 | TURKEY<br>Marmara University Pendik Education and Training Hospital's Child and Adolescent Psychiatry Clinic | 0 | Included in papers 1 (epidemiology) & 2 (mental health) |
| 4 | Alastanos, J. N., & Mullen, S. (2017). Psychiatric admission in adolescent transgender patients: A case series. The Mental Health Clinician, 7(4), 172–175. | 2017 | USA<br>'an inpatient psychiatry unit' | 3<br>Selected case series | All 5 participants were psychiatric inpatients within a 5 week period |
| 5 | Alberse, et al. (2019). Self-perception of transgender clinic referred gender diverse children and adolescents. Clinical Child Psychology and Psychiatry, 24(2), 388–401. | 2019 | NETHERLANDS<br>Center of Expertise on Gender Dysphoria (CEGD), Amsterdam | 1<br>Max age 18.03 | Poor self-perception common. NF perceive themselves more positively in general. |
| 6 | Alexander, G. M., & Peterson, B. S. (2004). Testing the prenatal hormone hypothesis of tic-related disorders: gender identity and gender role behavior. Development & Psychopathology, 16(2), 407–420. | 2004 | USA<br>Child Study Center of Yale University in New Haven, CT | 6 | Participants receiving treatment for Tourette Syndrome, not presenting for GD |
| 7 | Amir, H., Oren, A., Klochendler Frishman, E., Sapir, O., Shufaro, Y., Segev Becker, A.,. . . Ben-Haroush, A. (2020). Oocyte retrieval outcomes among adolescent transgender males. Journal of Assisted Reproduction & Genetics, 37(7), 1737–1744. | 2020 | ISRAEL<br>IVF Unit, Fertility Institute in Tel Aviv Sourasky Medical Center and IVF and the Infertility Unit, Helen Schneider Hospital for Women, Rabin Medical Center | 3 | Sample was 11 NF presenting specifically for fertility preservation |
| 8 | Amir, H., Yaish, I., Oren, A., Groutz, A., Greenman, Y., & Azem, F. (2020). Fertility preservation rates among transgender women compared with transgender men receiving comprehensive fertility counselling. Reproductive Biomedicine Online, 41 (3), 546–554. | 2020 | ISRAEL<br>Gender Dysphoria Clinic at Dana-Dwek Children's Hospital, Gender Clinic, Tel Aviv Sourasky Medical Center | 0 | Included in Paper 1 (epidemiology) |
| 9 | Anzani, A., Panfilis, C., Scandurra, C., & Prunas, A. (2020). Personality Disorders and Personality Profiles in a Sample of Transgender Individuals Requesting Gender-Affirming Treatments. International Journal of Environmental Research & Public Health, 17(5), 27. | 2020 | ITALY<br>gender clinic at Niguarda Ca' Granda Hospital, Milan | 3 | Very small sub-sample (n = 4) in adolescent age range |

*(Continued)*

**Table 4.** (Continued)

| ID | Reference | Date | Location & setting | Reason for exclusion | Notes |
|---|---|---|---|---|---|
| 10 | Arnoldussen, et al. (2019). Re-evaluation of the Dutch approach: are recently referred transgender youth different compared to earlier referrals? European Child and Adolescent Psychiatry. | 2019 | NETHERLANDS Center of Expertise on Gender Dysphoria (CEGD), Amsterdam | 1 Max age 18.08 | From 2000–2016, sharp increase in cases from 2012 to 2016; sharp uptick in NF relative to NM since 2013 |
| 11 | Avila, J. T., Golden, N. H., & Aye, T. (2019). Eating Disorder Screening in Transgender Youth. Journal of Adolescent Health, 65(6), 815–817. | 2019 | USA 'an academic multidisciplinary gender clinic' Stanford University School of Medicine* | 1, 2 No distinct data on adolescents in sample. | Most (63%) disclosed weight manipulation for gender-affirming purposes, including 11% of NF for menstrual suppression. |
| 12 | Barnard, E. P., Dhar, C. P., Rothenberg, S. S., Menke, M. N., Witchel, S. F., Montano, G. T.,. . . Valli-Pulaski, H. (2019). Fertility Preservation Outcomes in Adolescent and Young Adult Feminizing Transgender Patients. Pediatrics, 144(3). | 2019 | USA Magee-Womens Research Institute, Pittsburgh* | 1, 3 | Very small subsample (n = 11); only 4 in adolescence at time of assessment (consultation). |
| 13 | Becerra-Culqui, T. A., Liu, Y., Nash, R., Cromwell, L., Flanders, W. D., Getahun, D.,. . . Goodman, M. (2018). Mental health of transgender and gender nonconforming youth compared with their peers. Pediatrics, 141 (5) (no pagination)(e20173845). | 2018 | USA California and Georgia, US, Kaiser-Permanente records | 0 | Included in papers 1 (epidemiology) & 2 (mental health) |
| 14 | Becerra-Fernández, A., Rodríguez-Molina, J., Ly-Pen, D., Asenjo-Araque, N., Lucio-Pérez, M., Cuchí-Alfaro, M.,. . . Aguilar-Vilas, M. V. (2017). Prevalence, Incidence, and Sex Ratio of Transsexualism in the Autonomous Region of Madrid (Spain) According to Healthcare Demand. Archives of Sexual Behavior, 46(5), 1307–1312. doi:10.1007/s10508-017-0955-z | 2017 | SPAIN Gender Identity Unit, Madrid | 1 No separate data on adolescents (includes ≥18 yrs) | Higher prevalence rate than other countries: attributed to easily accessible services and positive social and legal climate in Spain. |
| 15 | Bechard, M., VanderLaan, D. P., Wood, H., Wasserman, L., & Zucker, K. J. (2017). Psychosocial and Psychological Vulnerability in Adolescents with Gender Dysphoria: A "Proof of Principle" Study. Journal of Sex & Marital Therapy, 43(7), 678–688. | 2017 | CANADA Gender Identity Service, CYFS, Toronto | 1 No separate data on adolescents (includes ≥18 yrs) | Mean of 5.56/13 'psychological vulnerability factors' amongst sample. |
| 16 | Becker, I., Auer, M., Barkmann, C., Fuss, J., Moller, B., Nieder, T. O.,. . . Richter-Appelt, H. (2018). A Cross-Sectional Multicenter Study of Multidimensional Body Image in Adolescents and Adults with Gender Dysphoria Before and After Transition-Related Medical Interventions. Archives of Sexual Behavior, 47(8), 2335–2347. | 2018 | GERMANY Department of Child and Adolescent Psychiatry, Psychotherapy, and Psycho-Somatics, University Medical Centre, Hamburg | 1 No separate data on adolescents (includes ≥18 yrs) | Body image generally poor; some (but not all) aspects of poor body image improved with intervention. |
| 17 | Biggs, M. (2020). Gender Dysphoria and Psychological Functioning in Adolescents Treated with GnRHa: Comparing Dutch and English Prospective Studies. Archives of Sexual Behavior, 49(7), 2231–2236. | 2020 | Secondary data: NETHERLANDS and UK | 5 No original data | Useful critique of existing data: compares Dutch and English samples |

(*Continued*)

**Table 4.** (Continued)

| ID | Reference | Date | Location & setting | Reason for exclusion | Notes |
|----|-----------|------|--------------------|-----------------------|-------|
| 18 | Bonifacio, J. H., Maser, C., Stadelman, K., & Palmert, M. (2019). Management of gender dysphoria in adolescents in primary care. Cmaj, 191(3), E69-E75. | 2019 | N/A–review paper (Authors based in Toronto, CANADA) | 5 | Increase in cases will mean primary care services have to be prepared |
| 19 | Bradley, S. J. (1978). Gender identity problems of children and adolescents: The establishment of a special clinic. The Canadian Psychiatric Association Journal / La Revue de l'Association des psychiatres du Canada, 23(3), 175–183. | 1978 | CANADA Toronto, southwestern Ontario | 3 Small N. | No objective data–clinical impressions only. Interesting early paper on GD and Toronto clinical service. |
| 20 | Brocksmith, V. M., Alradadi, R. S., Chen, M., & Eugster, E. A. (2018). Baseline characteristics of gender dysphoric youth. *Journal of Pediatric Endocrinology and Metabolism*, 31(12), 1367–1369. | 2018 | USA pediatric endocrine clinic, Riley Hospital for Children, Indianapolis | 1 No separate data on adolescents (includes ≥18 yrs) | High proportion of NF to NM 50% overweight / obese Anxiety higher in NF Some indication of adolescent-onset being more common in recent (post-2014) participants |
| 21 | Bui, H. N., Schagen, S. E. E., Klink, D. T., Delemarre-Van De Waal, H. A., Blankenstein, M. A., & Heijboer, A. C. (2013). Salivary testosterone in female-To-male transgender adolescents during treatment with intra-muscular injectable testosterone esters. Steroids, 78(1), 91–95. | 2013 | NETHERLANDS VU Medical Center, Amsterdam | 1, 3 | Technical paper focused on novel method of measuring salivary testosterone levels. |
| 22 | Burke, S. M., Kreukels, B. P. C., Cohen-Kettenis, P. T., Veltman, D. J., Klink, D. T., & Bakker, J. (2016). Male-typical visuospatial functioning in gynephilic girls with gender dysphoria—Organizational and activational effects of testosterone. Journal of Psychiatry and Neuroscience, 41(6), 395–404. | 2016 | NETHERLANDS Center of Expertise on Gender Dysphoria (CEGD), Amsterdam | 3 | Small selected sample, unlikely to be representative. Psychiatric disorder an exclusion criterion. |
| 23 | Butler, G., De Graaf, N., Wren, B., & Carmichael, P. (2018). Assessment and support of children and adolescents with gender dysphoria. *Archives of Disease in Childhood*, 103(7), 631–636. | 2018 | UK Gender Identity Development Service, Tavistock, London | 5 Commissioned review | Some data on age and gender |
| 24 | Calzo, J. P., & Blashill, A. J. (2018). Child Sexual Orientation and Gender Identity in the Adolescent Brain Cognitive Development Cohort Study. JAMA Pediatrics, 172(11), 1090–1092. | 2018 | USA San Diego (Adolescent Brain Cognitive Development (ABCD) study) | 2, 4 | Self-identification of sexuality and gender status, with very small N identifying as transgender. Sample aged 9–10 yrs. |
| 25 | Chen, D., Simons, L., Johnson, E. K., Lockart, B. A., & Finlayson, C. (2017). Fertility Preservation for Transgender Adolescents. Journal of Adolescent Health, 61(1), 120–123. | 2017 | USA Gender & Sex Development Program (GSDP), Ann & Robert H. Lurie Children's Hospital of Chicago | 3 | Very small sample. 11/13 were adolescents. Sample was 11 adolescents presenting specifically for fertility preservation. |
| 26 | Chiniara, L. N., Bonifacio, H. J., & Palmert, M. R. (2018). Characteristics of adolescents referred to a gender clinic: Are youth seen now different from those in initial reports? Hormone Research in Paediatrics, 89(6), 434–441. | 2018 | CANADA Transgender Youth Clinic (TYC), The Hospital for Sick Children, Toronto | 0 | Included in papers 1 (epidemiology) & 2 (mental health) |
| 27 | Chodzen, G., Hidalgo, M. A., Chen, D., & Garofalo, R. (2019). Minority Stress Factors Associated With Depression and Anxiety Among Transgender and Gender-Nonconforming Youth. *Journal of Adolescent Health*, 64(4), 467–471. | 2019 | USA Division of Adolescent Medicine, Ann & Robert H. Lurie Children's Hospital of Chicago* | 2 Sample self-identified as transgender and gender-nonconforming (TGNC) | High levels of anxiety and depression. |

(*Continued*)

**Table 4.** (Continued)

| ID | Reference | Date | Location & setting | Reason for exclusion | Notes |
|---|---|---|---|---|---|
| 28 | Clark, T. C., Lucassen, M. F. G., Bullen, P., Denny, S. J., Fleming, T. M., Robinson, E. M., & Rossen, F. V. (2014). The health and well-being of transgender high school students: Results from the New Zealand adolescent health survey (youth'12). *Journal of Adolescent Health*, *55*(1), 93–99. | 2014 | NEW ZEALAND<br>National population-based survey | 2<br>Survey data. Participants self-identify as transgender. | Adolescents identifying as transgender have considerable health and wellbeing needs relative to their peers. |
| 29 | Cohen, L., De Ruiter, C., Ringelberg, H., & Cohen-Kettenis, P. T. (1997). Psychological functioning of adolescent transsexuals: Personality and psychopathology. Journal of Clinical Psychology, 53(2), 187–196. | 1997 | NETHERLANDS<br>VU Medical Center, Amsterdam | 1 | Mean age 17.2±1.81<br>Rorschach methodology–not useful for our review |
| 30 | Cohen-Kettenis, P. T., & Van Goozen, S. H. M. (1997). Sex reassignment of adolescent transsexuals: A follow-up study. *Journal of the American Academy of Child and Adolescent Psychiatry, 36*(2), 263–271. | 1997 | NETHERLANDS<br>University Medical Centre, Utrecht (moved to VUMC / CEGD in 2002). | 1, 3 | GD resolved at post-surgery follow-up. No expression of regret. |
| 31 | Cohen-Kettenis, P. T., & Van Goozen, S. H. M. (2002). Adolescents who are eligible for sex reassignment surgery: Parental reports of emotional and behavioural problems. Clinical Child Psychology and Psychiatry, 7(3), 412–422. | 2002 | NETHERLANDS<br>University Medical Centre, Utrecht (moved to VUmc / CEGD in 2002) | 0 | Included in papers 1 (epidemiology) & 2 (mental health) |
| 32 | Coolidge, F. L., Thede, L. L., & Young, S. E. (2002). The heritability of gender identity disorder in a child and adolescent twin sample. Behavior Genetics, 32(4), 251–257. | 2002 | USA<br>Colorado Springs, Colorado | 2 | Twin methdology to investigate heritability. |
| 33 | Day, J. K., Fish, J. N., Perez-Brumer, A., Hatzenbuehler, M. L., & Russell, S. T. (2017). Transgender Youth Substance Use Disparities: Results From a Population-Based Sample. *Journal of Adolescent Health*, *61*(6), 729–735. doi:10.1016/j.jadohealth.2017.06.024 | 2017 | USA<br>2013–2015<br>Biennial Statewide California Student Survey | 2<br>Survey data. Participants self-identify as transgender. | Transgender youth at increased risk for substance misuse. Some psychosocial factors may mediate this. |
| 34 | de Graaf, et al. (2018). Sex ratio in children and adolescents referred to the Gender Identity Development Service in the UK (2009–2016). Archives of Sexual Behavior, 47(5), 1301–1304. doi:10.1007/s10508-018-1204-9 | 2018 | UK<br>Gender Identity Development Service, Tavistock, London | 2<br>Referrals only–no GD clinical verification | Significant increases year-on-year, from only 39 adolescent referrals in 2009 to almost 1500 in 2016; average increase rate of referrals higher in NF. |
| 35 | de Graaf, N. M., Carmichael, P., Steensma, T. D., & Zucker, K. J. (2018). Evidence for a Change in the Sex Ratio of Children Referred for Gender Dysphoria: Data From the Gender Identity Development Service in London (2000–2017). Journal of Sexual Medicine, 15(10), 1381–1383. | 2018 | UK<br>Gender Identity Development Service, Tavistock, London | 4 | NM referred at younger age than NF. Recent increases have higher proportion NF (as observed in adolescent samples). |

*(Continued)*

**Table 4.** (*Continued*)

| ID | Reference | Date | Location & setting | Reason for exclusion | Notes |
|----|-----------|------|--------------------|-----------------------|-------|
| 36 | de Graaf, N. M., Cohen-Kettenis, P. T., Carmichael, P., de Vries, A. L. C., Dhondt, K., Laridaen, J.,. . . Steensma, T. D. (2018). Psychological functioning in adolescents referred to specialist gender identity clinics across Europe: A clinical comparison study between four clinics. European Child & Adolescent Psychiatry, 27(7), 909–919. doi:10.1007/s00787-017-1098-4 | 2018 | MULTI (a) Center of Expertise on Gender Dysphoria (CEGD), Amsterdam, NETHERLANDS; (b) Pediatric Gender Clinic, Ghent University Hospital, BELGIUM; (c) Department of Child and Adolescent Psychiatry, University Hospital of Psychiatry Zurich, SWITZERLAND; (d) Gender Identity Development Service, Tavistock and Portman NHS Foundation Trust, London, UK | 0 | Included in papers 1 (epidemiology) & 2 (mental health) |
| 37 | de Graaf, N. M., Manjra, II, Hames, A., & Zitz, C. (2019). Thinking about ethnicity and gender diversity in children and young people. Clinical Child Psychology & Psychiatry, 24(2), 291–303. | 2019 | UK Gender Identity Development Service, Tavistock, London | 2 Referrals only–no GD clinical verification | Black and minority ethnic groups were underrepresented. |
| 38 | De Pedro, K. T., Gilreath, T. D., Jackson, C., & Esqueda, M. C. (2017). Substance Use Among Transgender Students in California Public Middle and High Schools. The Journal of school health, 87(5), 303–309. | 2017 | US 2013–2015 California Healthy Kids Survey (CHKS) | 2 Survey. Self-identified as transgender. | Transgender youth at increased risk for substance misuse. |
| 39 | De Vries, A. L. C., Noens, I. L. J., Cohen-Kettenis, P. T., Van Berckelaer-Onnes, I. A., & Doreleijers, T. A. (2010). Autism spectrum disorders in gender dysphoric children and adolescents. Journal of Autism and Developmental Disorders, 40(8), 930–936. | 2010 | NETHERLANDS VU Medical Center, Amsterdam | 2 Not all participants diagnosed | Indication of association between GID and ASD |
| 40 | de Vries, A. L., Doreleijers, T. A., Steensma, T. D., & Cohen-Kettenis, P. T. (2011). Psychiatric comorbidity in gender dysphoric adolescents. Journal of child psychology and psychiatry, and allied disciplines, 52(11), 1195–1202. | 2011 | NETHERLANDS VU Medical Center, Amsterdam | 0 | Included in papers 1 (epidemiology) & 2 (mental health) |
| 41 | De Vries, A. L. C., McGuire, J. K., Steensma, T. D., Wagenaar, E. C. F., Doreleijers, T. A. H., & Cohen-Kettenis, P. T. (2014). Young adult psychological outcome after puberty suppression and gender reassignment. Pediatrics, 134(4), 696–704. | 2014 | NETHERLANDS VU Medical Center, Amsterdam | 0 | Included in paper 1 (epidemiology) |
| 42 | de Vries, A. L. C., Steensma, T. D., Cohen-Kettenis, P. T., VanderLaan, D. P., & Zucker, K. J. (2016). Poor peer relations predict parent- and self-reported behavioral and emotional problems of adolescents with gender dysphoria: a cross-national, cross-clinic comparative analysis. European Child and Adolescent Psychiatry, 25(6), 579–588. | 2016 | NETHERLANDS VU Medical Center, Amsterdam | 0 | Included in papers 1 (epidemiology) & 2 (mental health) |

(*Continued*)

**Table 4.** (Continued)

| ID | Reference | Date | Location & setting | Reason for exclusion | Notes |
|---|---|---|---|---|---|
| 43 | Delahunt, J. W., Denison, H. J., Sim, D. A., Bullock, J. J., & Krebs, J. D. (2018). Increasing rates of people identifying as transgender presenting to Endocrine Services in the Wellington region. *New Zealand Medical Journal, 131*(1468), 33–42. | 2018 | NEW ZEALAND Wellington Endocrine Service for Capital & Coast District Health Board | 1 No separate data on adolescents (includes ≥18 yrs) | Observed increase in referrals for people under 30, also increase in requests for female to male transition. |
| 44 | Drummond, K. D., Bradley, S. J., Peterson-Badali, M., VanderLaan, D. P., & Zucker, K. J. (2018). Behavior Problems and Psychiatric Diagnoses in Girls with Gender Identity Disorder: A Follow-Up Study. Journal of Sex & Marital Therapy, 44(2), 172–187. | 2018 | CANADA Gender Identity Service, Center for Addiction and Mental Health, Toronto | 1, 6 Included some DSD; No separate data on adolescents | 12% (n = 3) of those referred in childhood showed persistent GID in adolescence / adulthood. Some indication of psychiatric vulnerability at follow-up, but a lot of variability. |
| 45 | Drummond, K. D., Bradley, S. J., Peterson-Badali, M., & Zucker, K. J. (2008). A follow-up study of girls with gender identity disorder. Developmental Psychology, 44(1), 34–45. | 2008 | CANADA Gender Identity Service, Center for Addiction and Mental Health, Toronto | 1 Follow-up took place from age 17; small n under 18 (and no distinct data reported) | As above |
| 46 | Durwood, L., McLaughlin, K. A., & Olson, K. R. (2017). Mental health and self-worth in socially transitioned transgender youth. *Journal of the American Academy of Child & Adolescent Psychiatry*, 56(2), 116–123. doi:10.1016/j.jaac.2016.10.016 | 2017 | NORTH AMERICA Multisite survey–TransYouth Project | 2 Survey data. Participants self-identify as transgender. | Socially transitioned young people had MH no worse than others their age |
| 47 | Edwards-Leeper, L., Feldman, H. A., Lash, B. R., Shumer, D. E., & Tishelman, A. C. (2017). Psychological profile of the first sample of transgender youth presenting for medical intervention in a US pediatric gender center. Psychology of Sexual Orientation and Gender Diversity, 4(3), 374–382. doi:10.1037/sgd0000239 | 2017 | USA The Gender Management Service (GeMS) program at Boston Children's Hospital | 0 | Included in papers 1 (epidemiology) & 2 (mental health) |
| 48 | Feder, S., Isserlin, L., Seale, E., Hammond, N., & Norris, M. L. (2017). Exploring the association between eating disorders and gender dysphoria in youth. Eating Disorders, 25(4), 310–317. | 2017 | CANADA Gender Diversity Clinic at a Canadian tertiary pediatric care hospital in Ottawa, Ontario | 0 | Included in papers 1 (epidemiology) & 2 (mental health) |
| 49 | Fisher, A. D., Ristori, J., Castellini, G., Sensi, C., Cassioli, E., Prunas, A.,. . . Maggi, M. (2017). Psychological characteristics of Italian gender dysphoric adolescents: a case-control study. Journal of Endocrinological Investigation, 40(9), 953–965. | 2017 | ITALY The Sexual Medicine and Andrology Unit of the University of Florence and the Gender Clinics of Rome, Milan, and Naples University Hospitals | 0 | Included in papers 1 (epidemiology) & 2 (mental health) |
| 50 | Getahun, D., Nash, R., Flanders, W. D., Baird, T. C., Becerra-Culqui, T. A., Cromwell, L.,. . . Goodman, M. (2018). Cross-sex Hormones and Acute Cardiovascular Events in Transgender Persons: A Cohort Study. Annals of Internal Medicine, 169(4), 205–213. doi:10.7326/M17-2785 | 2018 | USA Kaiser Permanente sites in Georgia, northern California, southern California | 1 Adults only | Indication of increased risk of cardiovascular events in transfeminine participants. |

(*Continued*)

**Table 4.** (*Continued*)

| ID | Reference | Date | Location & setting | Reason for exclusion | Notes |
|----|-----------|------|--------------------|-----------------------|-------|
| 51 | Handler, T., Hojilla, J. C., Varghese, R., Wellenstein, W., Satre, D. D., & Zaritsky, E. (2019). Trends in Referrals to a Pediatric Transgender Clinic. Pediatrics, 144(5), 11. | 2019 | USA<br>Kaiser Permanente Northern California health system | 2<br>No clinical verification of GD / GID. | Observed increase in referrals in recent years. Large proportion identify as transmasculine. Treatment needs varied by age group. |
| 52 | Hannema, S. E., Schagen, S. E. E., Cohen-Kettenis, P. T., & Delemarre-Van De Waal, H. A. (2017). Efficacy and safety of pubertal induction using 17beta-estradiol in transgirls. *Journal of Clinical Endocrinology and Metabolism*, *102*(7), 2356–2363. | 2017 | NETHERLANDS<br>Centre of Expertise on Gender Dysphoria (CEGD), Amsterdam | 1<br>No separate data on adolescents (includes ≥18 yrs) | Estradiol effective for pubertal induction in NM. |
| 53 | Heard, J., Morris, A., Kirouac, N., Ducharme, J., Trepel, S., & Wicklow, B. (2018). Gender dysphoria assessment and action for youth: Review of health care services and experiences of trans youth in Manitoba. Paediatrics & Child Health (1205–7088), 23(3), 179–184. doi:10.1093/pch/pxx156 | 2018 | CANADA<br>Manitoba Gender Dysphoria Assessment and Action for Youth (GDAAY) program | 0 | Included in papers 1 (epidemiology) & 2 (mental health) |
| 54 | Holt, V., Skagerberg, E., & Dunsford, M. (2016). Young people with features of gender dysphoria: Demographics and associated difficulties. Clinical Child Psychology and Psychiatry, 21(1), 108–118. | 2016 | UK<br>Gender Identity Development Service, Tavistock & Portman, London | 0 | Included in papers 1 (epidemiology) & 2 (mental health) |
| 55 | Hughes, S. K., VanderLaan, D. P., Blanchard, R., Wood, H., Wasserman, L., & Zucker, K. J. (2017). The Prevalence of Only-Child Status Among Children and Adolescents Referred to a Gender Identity Service Versus a Clinical Comparison Group. *Journal of Sex & Marital Therapy*, *43*(6), 586–593. | 2017 | CANADA<br>Gender Identity Service, CYFS, Toronto | 2, 4<br>Mean age of all groups <12 yrs. No apparent clinical verification of GD / GID. | Prevalence of only-child status not elevated in gender-referred children (compared to other clinical populations). |
| 56 | Janssen, A., Huang, H., & Duncan, C. (2016). Gender Variance among Youth with Autism Spectrum Disorders: A Retrospective Chart Review. *Transgender Health*, *1*(1), 63–68. | 2016 | USA<br>New York University Child Study Center | 2<br>Derived from clinical ASD sample.<br>No clinical verification of GD / GID. | Participants with ASD diagnosis more likely to report gender variance on CBCL then CBCL normative sample. |
| 57 | Jarin, J., Pine-Twaddell, E., Trotman, G., Stevens, J., Conard, L. A., Tefera, E., & Gomez-Lobo, V. (2017). Cross-sex hormones and metabolic parameters in adolescents with gender dysphoria. Pediatrics, 139 (5) (no pagination) (e20163173). | 2017 | USA<br>Multi-site.<br>MedStar Washington Hospital Center and<br>Children's National Medical Center (both, Washington, DC).<br>University of Maryland Medical Center, Baltimore.<br>Cincinnati Children's Hospital Medical Center, Ohio. | 1<br>No separate data on adolescents (includes ≥18 yrs) | Testosterone use associated with increased hemoglobin and hematocrit, increased BMI, and lowered high-density lipoprotein levels. No significant change in those taking estrogen. |
| 58 | Kaltiala, R., Bergman, H., Carmichael, P., de Graaf, N. M., Egebjerg Rischel, K., Frisen, L.,. . . Waehre, A. (2020). Time trends in referrals to child and adolescent gender identity services: a study in four Nordic countries and in the UK. Nordic Journal of Psychiatry, 74 (1), 40–44. | 2020 | DENMARK, FINLAND, NORWAY, SWEDEN, & the UK | 2<br>No clinical verification of GD / GID. | Comprehensive overview of referrals in 5 countries. Same pattern of increase, especially in NF, as in included papers. |

(*Continued*)

**Table 4.** (Continued)

| ID | Reference | Date | Location & setting | Reason for exclusion | Notes |
|---|---|---|---|---|---|
| 59 | Kaltiala, R., Heino, E., Tyolajarvi, M., & Suomalainen, L. (2020). Adolescent development and psychosocial functioning after starting cross-sex hormones for gender dysphoria. Nordic Journal of Psychiatry, 74(3), 213–219. | 2020 | FINLAND<br>Tampere University Hospital, Department of Adolescent Psychiatry | 1<br>Mean age at assessment 18.1 | MH problems persisted during treatment–concluded that GD treatment not enough to address MH problems. |
| 60 | Kaltiala-Heino, R., Sumia, M., Tyolajarvi, M., & Lindberg, N. (2015). Two years of gender identity service for minors: Overrepresentation of natal girls with severe problems in adolescent development. Child and Adolescent Psychiatry and Mental Health, 9 (1) (no pagination)(9). | 2015 | FINLAND<br>Tampere University Hospital, Department of Adolescent Psychiatry | 0 | Included in papers 1 (epidemiology) & 2 (mental health) |
| 61 | Kaltiala-Heino, R., Tyolajarvi, M., & Lindberg, N. (2019). Sexual experiences of clinically referred adolescents with features of gender dysphoria. Clinical Child Psychology and Psychiatry, 24(2), 365–378. | 2019 | FINLAND<br>Tampere University Hospital, Department of Adolescent Psychiatry | 0 | Included in papers 1 (epidemiology) & 2 (mental health) |
| 62 | Kaltiala-Heino, R., Tyolajarvi, M., & Lindberg, N. (2019). Gender dysphoria in adolescent population: A 5-year replication study. *Clinical Child Psychology and Psychiatry*, *24*(2), 379–387. | 2019 | FINLAND<br>School survey in Tampere | 2<br>Survey data.<br>No clinical verification of GD / GID. | Apparent increase in likely clinically-significant GD in adolescent population (2013–2017). |
| 63 | Katz-Wise, S. L., Ehrensaft, D., Vetters, R., Forcier, M., & Austin, S. B. (2018). Family Functioning and Mental Health of Transgender and Gender-Nonconforming Youth in the Trans Teen and Family Narratives Project. Journal of sex research, 55(4–5), 582–590. | 2018 | USA<br>New England region–survey from range of services / organisations | 2<br>Survey data.<br>No clinical verification of GD / GID. | MH concerns reported. Better family functioning (from young person's perspective) associated with better MH outcomes. |
| 64 | Khatchadourian, K., Amed, S., & Metzger, D. L. (2014). Clinical management of youth with gender dysphoria in Vancouver. *Journal of Pediatrics*, *164*(4), 906–911. doi:10.1016/j.jpeds.2013.10.068 | 2014 | CANADA<br>British Columbia<br>Children's Hospital Transgender Program, Vancouver | 1<br>No separate data on adolescents (includes ≥18 yrs) | Median age at initiation of testosterone NF 17.3 years<br>(range 13.7–19.8 years); median age at initiation of estrogen in NM 17.9 years (range 13.3–22.3 years). Intervnention appropriate in selected individuals with relevant clinical support. |
| 65 | Klein, D. A., Roberts, T. A., Adirim, T. A., Landis, C. A., Susi, A., Schvey, N. A., & Hisle-Gorman, E. (2019). Transgender Children and Adolescents Receiving Care in the US Military Health Care System. JAMA Pediatrics. | 2019 | USA<br>Military Health System Data Repository | 1<br>No separate data on adolescents (includes ≥18 yrs) | Increase in service use 2010–2017. Prescriptions increased with higher parental rank. |
| 66 | Kolbuck, V. D., Muldoon, A. L., Rychlik, K., Hidalgo, M. A., & Chen, D. (2019). Psychological functioning, parenting stress, and parental support among clinic-referred prepubertal gender-expansive children. *Clinical Practice in Pediatric Psychology*, *7*(3), 254–266. doi:10.1037/cpp0000293 | 2019 | USA<br>Division of Adolescent Medicine, Ann & Robert H. Lurie Children's Hospital of Chicago* | 4<br>Sample <12 yrs. | Association between GD symptoms in ADHD (hyperactive-impulsive) and CD where parenting stress high. |

(*Continued*)

**Table 4.** (Continued)

| ID | Reference | Date | Location & setting | Reason for exclusion | Notes |
|----|-----------|------|--------------------|-----------------------|-------|
| 67 | Kuper, L. E., Lindley, L., & Lopez, X. (2019). Exploring the gender development histories of children and adolescents presenting for gender affirming medical care. Clinical Practice in Pediatric Psychology, 7(3), 217–228. | 2019 | USA<br>Gender Education and Care Interdisciplinary Support program, Texas | 0 | Included in papers 1 (epidemiology) & 2 (mental health) |
| 68 | Kuper, L. E., Mathews, S., & Lau, M. (2019). Baseline Mental Health and Psychosocial Functioning of Transgender Adolescents Seeking Gender-Affirming Hormone Therapy. Journal of developmental and behavioral pediatrics: JDBP., 03. | 2019 | USA<br>Gender Education and Care Interdisciplinary Support program, Texas | 0 | Included in papers 1 (epidemiology) & 2 (mental health) |
| 69 | Lawlis, S. M., Donkin, H. R., Bates, J. R., Britto, M. T., & Conard, L. A. E. (2017). Health Concerns of Transgender and Gender Nonconforming Youth and Their Parents Upon Presentation to a Transgender Clinic. Journal of Adolescent Health, 61(5), 642–648. doi:10.1016/j.jadohealth.2017.05.025 | 2017 | USA<br>'a transgender clinic at a large tertiary pediatric hospital in the Midwest.'<br>Oklahoma University Children's Hospital* | 2 | 66.1% attending for first appointment NF |
| 70 | Levitan, N., Barkmann, C., Richter-Appelt, H., Schulte-Markwort, M., & Becker-Hebly, I. (2019). Risk factors for psychological functioning in German adolescents with gender dysphoria: poor peer relations and general family functioning. European Child and Adolescent Psychiatry. | 2019 | GERMANY<br>Hamburg Gender Identity Service for Children and Adolescents | 0 | Included in papers 1 (epidemiology) & 2 (mental health) |
| 71 | Lobato, M. I., Koff, W. J., Schestatsky, S. S., Chaves, C. P. V., Petry, A., Crestana, T.,... Henriques, A. A. (2007). Clinical characteristics, psychiatric comorbidities and sociodemographic profile of transsexual patients from an outpatient clinic in Brazil. International Journal of Transgenderism, 10(2), 69–77. doi:10.1080/15532730802175148 | 2007 | BRAZIL<br>Hospital de Clínicas de Porto Alegre | 1 | 42.7% had at least one psychiatric comorbidity |
| 72 | Lothstein, L. M. (1980). The adolescent gender dysphoric patient: An approach to treatment and management. Journal of Pediatric Psychology, 5(1), 93–109. | 1980 | USA<br>Case Western<br>Reserve University (CWRU)<br>Gender Identity Clinic, Cleveland, Ohio | 1, 3 | Case series from over 40 years ago |
| 73 | Lynch, M. M., Khandheria, M. M., & Meyer, W. J., III. (2015). Retrospective study of the management of childhood and adolescent gender identity disorder using medroxyprogesterone acetate. International Journal of Transgenderism, 16(4), 201–208. doi:10.1080/15532739.2015.1080649 | 2015 | USA<br>Gender<br>Identity Clinic, University of Texas Medical Branch | 3<br>Case series | Medroxyprogesterone Acetate found to be effective and low-cost oral alternative to injectable or implant GnRH analogues. Response to treatment and compliance were favourable. |

*(Continued)*

**Table 4.** (*Continued*)

| ID | Reference | Date | Location & setting | Reason for exclusion | Notes |
|---|---|---|---|---|---|
| 74 | Mahfouda, S., Panos, C., Whitehouse, A. J. O., Thomas, C. S., Maybery, M., Strauss, P.,. . . Lin, A. (2019). Mental Health Correlates of Autism Spectrum Disorder in Gender Diverse Young People: Evidence from a Specialised Child and Adolescent Gender Clinic in Australia. Journal of Clinical Medicine, 8(10), 20. | 2019 | AUSTRALIA<br>Gender Diversity Service (GDS), Perth Children's Hospital.<br>GENTLE cohort study | 0 | Included in papers 1 (epidemiology) & 2 (mental health) |
| 75 | Manners, P. J. (2009). Gender identity disorder in adolescence: A review of the literature. Child and Adolescent Mental Health, 14(2), 62–68. | 2009 | UK (review)<br>Salomons Clinical Psychology Training Program, Canterbury Christ Church University | 5<br>Review | Review now out of date |
| 76 | Matthews, T., Holt, V., Sahin, S., Taylor, A., & Griksaitis, D. (2019). Gender Dysphoria in looked-after and adopted young people in a gender identity development service. Clinical Child Psychology & Psychiatry, 24(1), 112–128. doi:10.1177/1359104518791657 | 2019 | UK<br>Gender Identity Development Service, Tavistock & Portman, London | 0 | Included in papers 1 (epidemiology) & 2 (mental health) |
| 77 | May, T., Pang, K., & Williams, K. J. (2017). Gender variance in children and adolescents with autism spectrum disorder from the National Database for Autism Research. *International Journal of Transgenderism*, *18*(1), 7–15. doi:10.1080/15532739.2016.1241976 | 2017 | USA<br>National Database for Autism Research | 2<br>Derived from clinical ASD sample.<br>No clinical verification of GD / GID. | Higher prevalence of gender variance in ASD sample compared to non-referred samples (but similar to other clinical samples). |
| 78 | Millington, K., Liu, E., & Chan, Y. M. (2019). The utility of potassium monitoring in gender-diverse adolescents taking spironolactone. *Journal of the Endocrine Society*, *3*(5), 1031–1038. | 2019 | USA<br>Gender Management Service Program, Boston Children's Hospital | 1<br>Sample likely to include those ≥18 yrs | Hyperkalemia in patients taking spironolactone for gender transition rare. Routine electrolyte monitoring may be unnecessary. |
| 79 | Millington, K., Schulmeister, C., Finlayson, C., Grabert, R., Olson-Kennedy, J., Garofalo, R.,. . . Chan, Y. M. (2020). Physiological and Metabolic Characteristics of a Cohort of Transgender and Gender-Diverse Youth in the United States. Journal of Adolescent Health, 67(3), 376–383. | 2020 | USA<br>Children's<br>Hospital Los Angeles/University of Southern California, Boston Children's Hospital/Harvard Medical School, the Ann & Robert H.<br>Lurie Children's Hospital of Chicago/Northwestern University, and the Benioff Children's Hospital/University of California San Francisco | 4<br>Sample 1 aged 8–14 (so mostly under 12s); sample 2 aged 12–20 (so included over 18s). | Description of baseline metabolic characteristics–will be useful to see cohorts followed up. |
| 80 | Moyer, D. N., Connelly, K. J., & Holley, A. L. (2019). Using the PHQ-9 and GAD-7 to screen for acute distress in transgender youth: findings from a pediatric endocrinology clinic. Journal of Pediatric Endocrinology & Metabolism, 32(1), 71–74. | 2019 | USA<br>'a pediatric endocrinology clinic', Portland, Oregon | 0 | Included in papers 1 (epidemiology) & 2 (mental health) |
| 81 | Munck, E. T. (2000). A retrospective study of adolescents visiting a Danish clinic for sexual disorders. International Journal of Adolescent Medicine and Health, 12(2–3), 215–222. doi:10.1515/IJAMH.2000.12.2–3.215 | 2000 | DENMARK<br>Sexological Clinic, Copenhagen University Hospital | 1<br>Mean age over 20 years | Description of cohort 1686–1995. Up to age 16, majority were NM. From age 17, majority were NF. |

(*Continued*)

**Table 4.** (Continued)

| ID | Reference | Date | Location & setting | Reason for exclusion | Notes |
|---|---|---|---|---|---|
| 82 | Nahata, L., Tishelman, A. C., Caltabellotta, N. M., & Quinn, G. P. (2017). Low Fertility Preservation Utilization Among Transgender Youth. Journal of Adolescent Health, 61(1), 40–44. | 2017 | USA<br>Division of Endocrinology, Department of Pediatrics, Nationwide Children's Hospital, The Ohio State University College of Medicine, Columbus, Ohio* | 5<br>Same sample already described (Nahata et al., 2017) | See epidemiological data in main review (Nahata et al., 2017). |
| 83 | Neyman, A., Fuqua, J. S., & Eugster, E. A. (2019). Bicalutamide as an Androgen Blocker With Secondary Effect of Promoting Feminization in Male-to-Female Transgender Adolescents. Journal of Adolescent Health, 64(4), 544–546. | 2019 | USA<br>Pediatric Endocrine Clinic, Riley Hospital for Children, Indiana | 1, 3<br>Where adolescent data separated, includes very small sample (case series) | Evidence that bicalutamide may be viable alternative to gonadotrophin-releasing hormone analogues in NM ready to transition |
| 84 | O'Bryan, J., Scribani, M., Leon, K., Tallman, N., Wolf-Gould, C., Wolf-Gould, C., & Gadomski, A. (2020). Health-related quality of life among transgender and gender expansive youth at a rural gender wellness clinic. Quality of Life Research, 29(6), 1597–1607. | 2020 | USA<br>The Gender Wellness Center (GWC) of the Bassett Health-care Network, New York | 1<br>Upper age limit 25 yrs. No meaningful separate data on those under 18 yrs. | Poor MH reported (relative to general population). Long term follow-up needed. |
| 85 | Olson, J., Schrager, S. M., Belzer, M., Simons, L. K., & Clark, L. F. (2015). Baseline Physiologic and Psychosocial Characteristics of Transgender Youth Seeking Care for Gender Dysphoria. Journal of Adolescent Health, 57(4), 374–380. doi:10.1016/j.jadohealth.2015.04.027 | 2015 | USA<br>Center for Transyouth Health and Development, Children's Hospital Los Angeles, California | 1 | Awareness of gender incongruity from young age (mean 8.3 yrs). Physiological characteristics within normal ranges. 35% experiencing depression; 51% contemplated suicide; 30% attempted suicide. |
| 86 | Olson-Kennedy, J., Okonta, V., Clark, L. F., & Belzer, M. (2018). Physiologic Response to Gender-Affirming Hormones Among Transgender Youth. Journal of Adolescent Health, 62(4), 397–401. | 2018 | USA<br>Center for Transyouth Health and Development, Children's Hospital Los Angeles, California | 1 | Use of gender affirming hormones not associated with clinically significant changes in metabolic parameters. May not need to frequently monitor transgender adolescents. |
| 87 | Olson-Kennedy, J., Warus, J., Okonta, V., Belzer, M., & Clark, L. F. (2018). Chest Reconstruction and Chest Dysphoria in Transmasculine Minors and Young Adults: Comparisons of Nonsurgical and Postsurgical Cohorts. JAMA Pediatrics, 172(5), 431–436. doi:10.1001/jamapediatrics.2017.5440 | 2018 | USA<br>Center for Transyouth Health and Development, Children's Hospital Los Angeles, California | 1 | Mean age at chest surgery 17.5 (2.4) years. 49% younger than 18 years. All postsurgical participants (n = 68) felt surgery had been a good decision. Loss of nipple sensation most common side-effect. |
| 88 | Olson-Kennedy, J., Chan, Y. M., Garofalo, R., Spack, N., Chen, D., Clark, L.,. . . Rosenthal, S. (2019). Impact of Early Medical Treatment for Transgender Youth: Protocol for the Longitudinal, Observational Trans Youth Care Study. JMIR Research Protocols, 8(7), e14434. | 2019 | USA<br>Children's Hospital Los Angeles/ University of Southern California, Boston Children's Hospital/ Harvard University, Lurie Children's Hospital of Chicago/ Northwestern University, Benioff Children's Hospital/ University of California San Francisco | 0 | Included in paper 1 (epidemiology) |

*(Continued)*

**Table 4.** (Continued)

| ID | Reference | Date | Location & setting | Reason for exclusion | Notes |
|----|-----------|------|--------------------|-----------------------|-------|
| 89 | Ospina, N. M. S., Maraka, S., Rodriguez-Gutierrez, R., Davidge-Pitts, C. J., Nippoldt, T. B., & Murad, M. H. (2016). Effect of sex steroids on the bone health of transgender individuals: A systematic review and meta-analysis. Endocrine Reviews. Conference: 98th Annual Meeting and Expo of the Endocrine Society, ENDO, 37(2 Supplement 1). | 2016 | International (researcher based in USA) | 5 Systematic review | Bone mineral density (lumbar spine) increased in NM 12–24 months after initiating feminising hormone therapy. No changes in NF with masculinising therapy. |
| 90 | Pakpoor, J., Wotton, C. J., Schmierer, K., Giovannoni, G., & Goldacre, M. J. (2016). Gender identity disorders and multiple sclerosis risk: A national record-linkage study. Multiple Sclerosis, 22(13), 1759–1762. | 2016 | UK English national Hospital Episode Statistics (HES) and mortality data | 5 No useable data for our questions | |
| 91 | Pang, K. C., de Graaf, N. M., Chew, D., Hoq, M., Keith, D. R., Carmichael, P., & Steensma, T. D. (2020). Association of Media Coverage of Transgender and Gender Diverse Issues With Rates of Referral of Transgender Children and Adolescents to Specialist Gender Clinics in the UK and Australia. JAMA Network Open, 3(7), e2011161. | 2020 | UK & Australia | 2 Data from referral only: GD not clinically verified. | Evidence of association between media coverage and number of new referrals to services. |
| 92 | Perez-Brumer, A., Day, J. K., Russell, S. T., & Hatzenbuehler, M. L. (2017). Prevalence and Correlates of Suicidal Ideation Among Transgender Youth in California: Findings From a Representative, Population-Based Sample of High School Students. Journal of the American Academy of Child & Adolescent Psychiatry, 56(9), 739–746. doi:10.1016/j.jaac.2017.06.010 | 2017 | USA California Healthy Kids Survey | 2 | Transgender youth had 2.99 higher odds of reporting past-year suicidal ideation than non-transgender youth. |
| 93 | Peterson, C. M., Matthews, A., Copps-Smith, E., & Conard, L. A. (2017). Suicidality, Self-Harm, and Body Dissatisfaction in Transgender Adolescents and Emerging Adults with Gender Dysphoria. Suicide & Life-Threatening Behavior, 47(4), 475–482. | 2017 | USA Cincinnati Children's Hospital Medical Center Transgender Clinic, Ohio | 1 | 30.3% transgender youth reported history of ≥1 suicide attempt; 41.8% history self-injury. Higher suicidality in NF than NM. |
| 94 | Quinn, V. P., Nash, R., Hunkeler, E., Contreras, R., Cromwell, L., Becerra-Culqui, T. A.,. . . Goodman, M. (2017). Cohort profile: Study of Transition, Outcomes and Gender (STRONG) to assess health status of transgender people. BMJ Open, 7(12), e018121. | 2017 | USA Kaiser-Permanente records, California and Georgia | 1 No useable data by age group | Useable data (proportion NM to NF) described in Becerra-Culqui et al. (2018) |
| 95 | Reisner, S. L., Biello, K. B., Hughto, J. M. W., Kuhns, L., Mayer, K. H., Garofalo, R., & Mimiaga, M. J. (2016). Psychiatric diagnoses and comorbidities in a diverse, multicity cohort of young transgender women: Baseline Findings from Project LifeSkills. JAMA Pediatrics, 170(5), 481–486. | 2016 | USA: Chicago and Boston–Project LifeSkills | 1, 2 No separate data on adolescents (includes ≥18 yrs). Current GD not an inclusion criterion. | 41.5% of sample of NM had 1 or more mental health or substance dependence diagnoses; 20% had 2 or more comorbid psychiatric diagnoses |

(*Continued*)

**Table 4.** (Continued)

| ID | Reference | Date | Location & setting | Reason for exclusion | Notes |
|---|---|---|---|---|---|
| 96 | Reisner, S. L., Vetters, R., Leclerc, M., Zaslow, S., Wolfrum, S., Shumer, D., & Mimiaga, M. J. (2015). Mental health of transgender youth in care at an adolescent Urban community health center: A matched retrospective cohort study. *Journal of Adolescent Health*, 56 (3), 274–279. | 2015 | US Sidney Borum Jr Health Center, Boston | 1 No separate data on adolescents (includes ≥18 yrs; mean age 19.6±3.0). | Increased risk of MH problems in transgender vs cisgender youth. No difference by natal gender. |
| 97 | Rider, G. N., Berg, D., Pardo, S. T., Olson-Kennedy, J., Sharp, C., Tran, K. M.,. . . Keo-Meier, C. L. (2019). Using the Child Behavior Checklist (CBCL) with transgender/gender nonconforming children and adolescents. Clinical Practice in Pediatric Psychology, 7(3), 291–301. doi:10.1037/cpp0000296 | 2019 | USA Trans Youth and Family Allies project (national) | 2 Survey | No significant impact in use of gendered scoring templates on CBCL |
| 98 | Roberts, A. L., Rosario, M., Slopen, N., Calzo, J. P., & Austin, S. B. (2013). Childhood gender nonconformity, bullying victimization, and depressive symptoms across adolescence and early adulthood: An 11-year longitudinal study. Journal of the American Academy of Child & Adolescent Psychiatry, 52(2), 143–152. doi:10.1016/j.jaac.2012.11.006 | 2013 | USA Growing Up Today Study (GUTS) (national) | 1, 2 Survey Age range 12–30 | Large longitudinal cohort. Association between gender nonconformity and depressive symptoms. |
| 99 | Röder, M., Barkmann, C., Richter-Appelt, H., Schulte-Markwort, M., Ravens-Sieberer, U., & Becker, I. (2018). Health-related quality of life in transgender adolescents: Associations with body image and emotional and behavioral problems. International Journal of Transgenderism, 19(1), 78–91. doi:10.1080/15532739.2018.1425649 | 2018 | GERMANY Hamburg Gender Identity Service for Children and Adolescents | 1 Upper age 18.2 | Health related quality of life (HRQoL) generally poorer in ttransgender adolescents vs normative scores. Body satisfaction and internalising problems significant predictors of HRQoL. |
| 100 | Schagen, S. E., Delemarre-van de Waal, H. A., Blanchard, R., & Cohen-Kettenis, P. T. (2012). Sibling sex ratio and birth order in early-onset gender dysphoric adolescents. Archives of Sexual Behavior, 41(3), 541–549. | 2012 | NETHERLANDS VU University Medical Center, Amsterdam (forerunner to CEGD) | 0 | Included in paper 1 (epidemiology) |
| 101 | Schagen, S. E. E., Cohen-Kettenis, P. T., Delemarre-van de Waal, H. A., & Hannema, S. E. (2016). Efficacy and Safety of Gonadotropin-Releasing Hormone Agonist Treatment to Suppress Puberty in Gender Dysphoric Adolescents. Journal of Sexual Medicine, 13(7), 1125–1132. | 2016 | NETHERLANDS Centre of Expertise on Gender Dysphoria (CEGD), Amsterdam | 1 Upper age 18.6 | Triptorelin effective in suppressing puberty. Routine monitoring of gonadotropins, sex steroids, creatinine, and liver function may not be necessary. |
| 102 | Schagen, et al. (2018). Changes in Adrenal Androgens During Puberty Suppression and Gender-Affirming Hormone Treatment in Adolescents With Gender Dysphoria. Journal of Sexual Medicine, 15(9), 1357–1363. | 2018 | NETHERLANDS Centre of Expertise on Gender Dysphoria (CEGD), Amsterdam | 1 Max age 18.6 | No harmful effects of treatment of GnRHa and gender affirming hormone treatment on adrenal androgen levels were found during approximately 4 years of follow-up. |

(*Continued*)

**Table 4.** (*Continued*)

| ID | Reference | Date | Location & setting | Reason for exclusion | Notes |
|---|---|---|---|---|---|
| 103 | Shields, J. P., Cohen, R., Glassman, J. R., Whitaker, K., Franks, H., & Bertolini, I. (2013). Estimating population size and demographic characteristics of lesbian, gay, bisexual, and transgender youth in middle school. Journal of Adolescent Health, 52(2), 248–250. | 2013 | USA<br>Youth Risk Behavior Survey (YRBS), San Francisco, California | 2<br>Survey | 1.3% of middle school students identified as transgender |
| 104 | Shumer, D. E., Reisner, S. L., Edwards-Leeper, L., & Tishelman, A. (2016). Evaluation of Asperger Syndrome in Youth Presenting to a Gender Dysphoria Clinic. *LGBT Health*, 3(5), 387–390. | 2016 | USA<br>Boston Children's Hospital | 1, 3<br>Sample included adults (max age 20 yrs).<br>Small sub-sample: only 6 aged 12–18 yrs ASQ>80 | 9/39 (23.1%) GD participants had indication of Asperger Syndrome. |
| 105 | Skagerberg, E., Di Ceglie, D., & Carmichael, P. (2015). Brief Report: Autistic Features in Children and Adolescents with Gender Dysphoria. *Journal of Autism and Developmental Disorders*, 45(8), 2628–2632. | 2015 | UK<br>Gender Identity Development Service, Tavistock, London | 2<br>No GD dx reported | Positive association between SRS scores and ASD symptoms. |
| 106 | Skagerberg, E., Parkinson, R., & Carmichael, P. (2013). Self-harming thoughts and behaviors in a group of children and adolescents with gender dysphoria. *International Journal of Transgenderism*, 14(2), 86–92. doi:10.1080/15532739.2013.817321 | 2013 | UK<br>Gender Identity Development Service, Tavistock, London | 2<br>No GD dx reported | 24% self-harmed, 14% had thoughts of self-harming, suicide attempts indicated in 10% prior to attending GIDS. Thoughts of self-harm more common in NM, actual self-harm more common in NF. |
| 107 | Skagerberg, E., Davidson, S., & Carmichael, P. (2013). Internalizing and externalizing behaviors in a group of young people with gender dysphoria. International Journal of Transgenderism, 14(3), 105–112. doi:10.1080/15532739.2013.822340 | 2013 | UK<br>Gender Identity Development Service (GIDS), Tavistock & Portman, London | 0 | Included in papers 1 (epidemiology) & 2 (mental health) |
| 108 | Smith, Y. L. S., Van Goozen, S. H. M., & Cohen-Kettenis, P. T. (2001). Adolescents with gender identity disorder who were accepted or rejected for sex reassignment surgery: A prospective follow-up study. *Journal of the American Academy of Child and Adolescent Psychiatry*, 40(4), 472–481. | 2001 | NETHERLANDS<br>University Medical Centre, Utrecht (moved to VUmc / CEGD in 2002). | 1<br>No separate data on adolescents (includes ≥18 yrs) | Group no longer GD after sex reassignment surgery. No one expressed regrets. Non-treated group showed some improvement in MH, but also had 'more dysfunctional psychological profile'. |
| 109 | Smith, Y. L. S., Van Goozen, S. H. M., Kuiper, A. J., & Cohen-Kettenis, P. T. (2005). Sex reassignment: Outcomes and predictors of treatment for adolescent and adult transsexuals. Psychological Medicine, 35(1), 89–99. | 2005 | NETHERLANDS<br>VU University<br>Medical Centre, Amsterdam (VUmc) or University Medical Centre, Utrecht (UMCU) | 1<br>No separate data on adolescents (includes ≥18 yrs) | Group no longer GD after sex reassignment surgery. |
| 110 | Sorbara, J. C., Chiniara, L. N., Thompson, S., & Palmert, M. R. (2020). Mental Health and Timing of Gender-Affirming Care. Pediatrics, 146(4), 10. | 2020 | CANADA<br>Transgender Youth Clinic (TYC), The Hospital for Sick Children, Toronto | 0 | Included in papers 1 (epidemiology) & 2 (mental health) |
| 111 | Spack, N. P., Edwards-Leeper, L., Feldman, H. A., Leibowitz, S., Mandel, F., Diamond, D. A., & Vance, S. R. (2012). Children and adolescents with gender identity disorder referred to a pediatric medical center. *Pediatrics*, 129(3), 418–425. | 2012 | USA<br>GeMS clinic, Endocrine Division, Children's Hospital Boston | 1<br>No separate data on adolescents (includes ≥18 yrs) | 44.3% had significant psychiatric history. Noted four-fold increase in presentations of GID following establishment of specialist service. |

(*Continued*)

**Table 4.** (*Continued*)

| ID | Reference | Date | Location & setting | Reason for exclusion | Notes |
|---|---|---|---|---|---|
| 112 | Steensma, T. D., & Cohen-Kettenis, P. T. (2015). More than two developmental pathways in children with gender dysphoria? *Journal of the American Academy of Child and Adolescent Psychiatry*, 54(2), 147–148. | 2015 | NETHERLANDS Centre of Expertise on Gender Dysphoria (CEGD), Amsterdam | 1 Letter to the editor: cannot determine if cohort already described in included papers. | Posits distinction between 'persisters' and 'persisters after interruption'. |
| 113 | Steensma, T. D., McGuire, J. K., Kreukels, B. P. C., Beekman, A. J., & Cohen-Kettenis, P. T. (2013). Factors associated with desistence and persistence of childhood gender dysphoria: A quantitative follow-up study. *Journal of the American Academy of Child and Adolescent Psychiatry*, 52(6), 582–590. | 2013 | NETHERLANDS Centre of Expertise on Gender Dysphoria (CEGD), Amsterdam | 1 No separate data on adolescents (includes ≥18 yrs) | Persistence of GD associated with early intensity of symptoms and being NF. Noted differing presentation by natal gender. |
| 114 | Steensma, T. D., Zucker, K. J., Kreukels, B. P. C., VanderLaan, D. P., Wood, H., Fuentes, A., & Cohen-Kettenis, P. T. (2014). Behavioral and emotional problems on the Teacher's Report Form: A cross-national, cross-clinic comparative analysis of gender dysphoric children and adolescents. *Journal of Abnormal Child Psychology*, 42(4), 635–647. doi:10.1007/s10802-013-9804-2 | 2014 | NETHERLANDS Centre of Expertise on Gender Dysphoria (CEGD), Amsterdam AND CANADA Gender Identity Service, CYFS, Toronto | 1 No separate data on adolescents (includes ≥18 yrs) (Adolescent subgroup likely to include those up to age 19–20, based on means / SDs given) | Teacher-reported emotional and behavioral problems greater in adolescents than in children. Internalising and externalising problems greater in NM than NF. Canadian sample had greater emotional and behavioural problems than Dutch sample. |
| 115 | Strang, J. F., Powers, M. D., Knauss, M., Sibarium, E., Leibowitz, S. F., Kenworthy, L.,. . . Anthony, L. G. (2018). "They Thought It Was an Obsession": Trajectories and Perspectives of Autistic Transgender and Gender-Diverse Adolescents. Journal of Autism and Developmental Disorders, 48(12), 4039–4055. | 2018 | USA Center for Neuroscience and Behavioral Medicine, Children's National Health System, Washington, DC* | 1 No separate data on adolescents (includes ≥18 yrs) | Useful qualitative study on young people's perspectives. No relevant data for our research questions. |
| 116 | Sumia, M., Lindberg, N., Tyolajarvi, M., & Kaltiala-Heino, R. (2016). Early pubertal timing is common among adolescent girl-to-boy sex reassignment applicants. European Journal of Contraception and Reproductive Health Care, 21(6), 483–485. | 2016 | FINLAND Tampere University Hospital, Tampere Helsinki University Hospital, Helsinki | 2 | GD in adolescence associated with early pubertal timing in NF |
| 117 | Sumia, M., Lindberg, N., Tyolajarvi, M., & Kaltiala-Heino, R. (2017). Current and recalled childhood gender identity in community youth in comparison to referred adolescents seeking sex reassignment. *Journal of Adolescence*, 56, 34–39. | 2017 | FINLAND School survey in Tampere & clinically referred population: Tampere and Helsinki | 1 No separate data on adolescents (includes ≥18 yrs) (Adolescent subgroup likely to include those 18+, based on means / SDs given) | Interesting exploration of gender identity in GD and community samples. No data directly relevant to our research questions. |
| 118 | Tollit, M. A., Pace, C. C., Telfer, M., Hoq, M., Bryson, J., Fulkoski, N.,. . . Pang, K. C. (2019). What are the health outcomes of trans and gender diverse young people in Australia? Study protocol for the Trans20 longitudinal cohort study. BMJ Open, 9(11), e032151. | 2019 | AUSTRALIA Royal Children's Hospital Gender Service (RCHGS), Melbourne | 5 Cohort description | Protocol paper only. |

(*Continued*)

**Table 4.** (Continued)

| ID | Reference | Date | Location & setting | Reason for exclusion | Notes |
|---|---|---|---|---|---|
| 119 | Twist, J., & de Graaf, N. M. (2019). Gender diversity and non-binary presentations in young people attending the United Kingdom's National Gender Identity Development Service. *Clinical Child Psychology and Psychiatry*, 24(2), 277–290. | 2019 | UK<br>Gender Identity Development Service, Tavistock, London | 2<br>No clinical verification of GD / GID–new questionnaire completed at presentation. | Useful in relation to prevalence of different types of gender self-identification at clinics |
| 120 | van der Miesen, A. I. R., Hurley, H., Bal, A. M., & de Vries, A. L. C. (2018). Prevalence of the Wish to be of the Opposite Gender in Adolescents and Adults with Autism Spectrum Disorder. *Archives of Sexual Behavior*, 47(8), 2307–2317. | 2018 | NETHERLANDS<br>Centre of Expertise on Gender Dysphoria (CEGD), Amsterdam | 2<br>No clinical verification of GD / GID–endorsement of single item on YSR only. | Significantly more adolescents (6.5%) with ASD endorsed item expressing wish to be the opposite gender compared to the general population (3–5%). NF endorsed more then NM. Adolescents with ASD who endorsed gender item had higher YSR scores (poorer MH). No association with any specific subdomain of ASD. |
| 121 | van der Miesen, A. I. R., de Vries, A. L. C., Steensma, T. D., & Hartman, C. A. (2018). Autistic Symptoms in Children and Adolescents with Gender Dysphoria. Journal of Autism and Developmental Disorders, 48(5), 1537–1548. | 2018 | NETHERLANDS<br>Center of Expertise on Gender Dysphoria (CEGD), Amsterdam | 0 | Included in papers 1 (epidemiology) & 2 (mental health) |
| 122 | van der Miesen, A. I. R., Steensma, T. D., de Vries, A. L. C., Bos, H., & Popma, A. (2020). Psychological Functioning in Transgender Adolescents Before and After Gender-Affirmative Care Compared With Cisgender General Population Peers. Journal of Adolescent Health, 66(6), 699–704. | 2020 | NETHERLANDS<br>Centre of Expertise on Gender Dysphoria (CEGD), Amsterdam | 2 (group 1)<br>1 (group 2) | Poor MH in referrals. MH in those receiving treatment was similar to general population sample. |
| 123 | Van Donge, N., Schvey, N. A., Roberts, T. A., & Klein, D. A. (2019). Transgender Dependent Adolescents in the U.S. Military Health Care System: Demographics, Treatments Sought, and Health Care Service Utilization. *Military medicine*, 184(5–6), e447-e454. | 2019 | USA<br>Transgender and gender-diverse clinic for children of military personnel | 1<br>No separate data on adolescents (includes ≥18 yrs) | Mean age at first gender-related visit 14.5 years (SD 3.2). History of self-harm (42%), suicidal ideation (70%), suicide attempt (21%), and psychiatric hospitalisation (33%). |
| 124 | Vlot, M. C., Klink, D. T., den Heijer, M., Blankenstein, M. A., Rotteveel, J., & Heijboer, A. C. (2017). Effect of pubertal suppression and cross-sex hormone therapy on bone turnover markers and bone mineral apparent density (BMAD) in transgender adolescents. *Bone*, 95, 11–19. | 2017 | NETHERLANDS<br>Centre of Expertise on Gender Dysphoria (CEGD), Amsterdam | 1<br>No separate data on adolescents (includes ≥18 yrs) | Suppressing puberty by GnRHa leads to a decrease of bone turnover markers (BTMs) in transgender adolescents, but added value of evaluating BTMs in transgender adolescents seems to be limited and requires further research. DXA-scans remain important in follow-up. |
| 125 | Wallien, M. S. C., & Cohen-Kettenis, P. T. (2008). Psychosexual outcome of gender-dysphoric children. *Journal of the American Academy of Child and Adolescent Psychiatry*, 47(12), 1413–1423. | 2008 | NETHERLANDS<br>VU University Medical Center (forerunner to CEGD), Amsterdam | 4<br>Onset <12 years | Most children with GD were not GD after puberty. Those with persistent GD had more intense GD in childhood than those desisting. |
| 126 | Wallien, M. S. C., Swaab, H., & Cohen-Kettenis, P. T. (2007). Psychiatric comorbidity among children with gender identity disorder. *Journal of the American Academy of Child and Adolescent Psychiatry*, 46(10), 1307–1314. | 2007 | NETHERLANDS<br>VU University Medical Center (forerunner to CEGD), Amsterdam | 4<br>all < 12 years | 52% of GID children had one or more other diagnoses. Internalising problems more common (37%) than externalising (23%). 31% of GID group had anxiety disorder. |

*(Continued)*

**Table 4.** (*Continued*)

| ID | Reference | Date | Location & setting | Reason for exclusion | Notes |
|---|---|---|---|---|---|
| 127 | Watson, R. J., Veale, J. F., & Saewyc, E. M. (2017). Disordered eating behaviors among transgender youth: Probability profiles from risk and protective factors. International Journal of Eating Disorders, 50(5), 515–522. doi:10.1002/eat.22627 | 2017 | CANADA<br>Canadian Trans Youth Health Survey (national) | 2<br>Survey | High rates of eating disorder behaviour among self-identified transgender youth. Risk for eating disordered behaviours linked to enacted stigma and violence exposure, and offset by social supports. |
| 128 | Wood, H., Sasaki, S., Bradley, S. J., Singh, D., Fantus, S., Owen-Anderson, A.,. . . Zucker, K. J. (2013). Patterns of referral to a gender identity service for children and adolescents (1976–2011): age, sex ratio, and sexual orientation. *Journal of Sex & Marital Therapy*, 39(1), 1–6. | 2013 | CANADA<br>Gender Identity Service, CYFS, Toronto | 1, 4<br>No separate data on adolescents (includes <12 yrs and ≥18 yrs) | Sharp increase in adolescent referrals in 2004–2007 time period (compared to 1976–2003), continued into 2008–2011 time block. NF exceeded NM in most recent (2008–2011) cohort. |
| 129 | Yadegarfard, M., Ho, R., & Bahramabadian, F. (2013). Influences on loneliness, depression, sexual-risk behaviour and suicidal ideation among Thai transgender youth. Culture, Health & Sexuality, 15(6), 726–737. doi:10.1080/13691058.2013.784362 | 2013 | THAILAND<br>Survey through range of organisations, via Rainbow Sky Association, Bangkok | 1, 2 | Education level (did not graduate high school) associated with less loneliness but more depression than those with some university credit. |
| 130 | Zou, Y., Szczesniak, R., Teeters, A., Conard, L. A. E., & Grossoehme, D. H. (2018). Documenting an epidemic of suffering: low health-related quality of life among transgender youth. Quality of Life Research, 27(8), 2107–2115. | 2018 | USA<br>Transgender Clinic of the Division of Adolescent and Transition Medicine, Cincinnati Children's Hospital Medical Center, Ohio | 1 | Transgender / gender non-conforming youth reported low health related quality of life across all domains. Most were significantly lower than healthy peers or peers with chronic diseases. |
| 131 | Zucker, K. J., Owen, A., Bradley, S. J., & Ameeriar, L. (2002). Gender-dysphoric children and adolescents: A comparative analysis of demographic characteristics and behavioral problems. *Clinical Child Psychology and Psychiatry*, 7(3), 398–411. doi:10.1177/1359104502007003007 | 2002 | CANADA<br>Gender Identity Service, CYFS, Toronto | 1, 4<br>No separate data on adolescents (includes <12 yrs and ≥18 yrs) | 84.7% of adolescents had CBCL sum score in clinical range (>90th centile). Scores strongly predicted by peer relations scale (i.e., poor peer relations predicted behavioural psychopathology). |
| 132 | Zucker, K. J., Bradley, S. J., Owen-Anderson, A., Kibblewhite, S. J., & Cantor, J. M. (2008). Is gender identity disorder in adolescents coming out of the closet? Journal of Sex and Marital Therapy, 34(4), 287–290. | 2008 | CANADA<br>Gender Identity Service, CYFS, Toronto | 1 | Same sample as Wood (2013) above. No new data relevant to our research questions. |
| 133 | Zucker, K. J., Bradley, S. J., Owen-Anderson, A., Kibblewhite, S. J., Wood, H., Singh, D., & Choi, K. (2012). Demographics, behavior problems, and psychosexual characteristics of adolescents with gender identity disorder or transvestic fetishism. Journal of Sex & Marital Therapy, 38(2), 151–189. | 2012 | CANADA<br>Gender Identity Service, CYFS, Toronto | 1 | Percentage of youth with CBCL and YSR total scores in clinical range was similar to non-GID referred comparison group, higher than non-referred comparison group. |

(*Continued*)

**Table 4.** (Continued)

| ID | Reference | Date | Location & setting | Reason for exclusion | Notes |
|---|---|---|---|---|---|
| 134 | Zucker, K. J., Bradley, S. J., Owen-Anderson, A., Singh, D., Blanchard, R., & Bain, J. (2010). Puberty-blocking hormonal therapy for adolescents with gender identity disorder: A descriptive clinical study. Journal of Gay & Lesbian Mental Health, 15(1), 58–82. doi:10.1080/19359705.2011.530574 | 2010 | CANADA Gender Identity Service, CYFS, Toronto | 1 | More likely to recommend puberty blockers for NF than NM, and less likely to recommend for young people with a lower YSR score. |

Key: * = derived from author's affiliation and description in paper

ADHD: Attention Deficit / Hyperactivity Disorder; ASD: Autism Spectrum Disorder; ASQ: Asperger Syndrome Quotient; CBCL: Child Behavior Check List; CD: Conduct Disorder; DXA: Dual-energy X-ray Absorptiometry; GD: Gender Dysphoria; GID: Gender Identity Disorder; GnRHa: Gonadotropin-releasing hormone agonist; MH: Mental Health; NM / NF: Natal Male / Natal Female; SRS: Social Responsiveness Scale; yrs: years.

Exclusion codes: 0: no data on treatment age or outcomes; 1: included ≥18 year olds–no distinct adolescent data; 2: non clinically-verified GD; 3: case study / series; 4: only included <12 year olds; 5: no original data; 6: non-GD population (e.g., LGBTQ); 7: conference proceedings

Four papers (Becker-Hebly et al., 2020; Costa et al., 2015; Kuper et al., 2020; Russell et al., 2020) reported that patients also had psychological interventions as part of their treatment. Costa et al. (2015) indicated that 7.5% of patients were referred to mental health services due to possible psychiatric comorbidities, which did not allow the start of medical intervention, and they received psychotherapy until they were eligible for PS treatment [43]. Becker-Hebly et al. (2020) indicated that the patients that were only undergoing psychosocial intervention (28%) were generally considered for medical treatment, but they needed to address psychological problems first [50].

Gender-affirming surgery (GAS) was not routinely offered in this young sample. Two papers reported cases obtaining GAS, with one reporting 15 NF adolescents obtaining mastectomy surgery, at an average age of 17.2 years old (range 15.2–18.7 years old) [49]. One centre in Germany mentions 11 patients obtaining GAS (14 mastectomy, 1 vaginoplasty) at a mean age of 18.0 years old (16.0–19.6 years old) [50]. Neither paper details how this surgery was obtained or where it was carried out. No other centres reported GAS in their populations.

**What outcomes are associated with treatment/s for GD in adolescence?.** Blood pressure, biochemistry and haematology

A range of cardiovascular and laboratory indicators of health were measured within the included studies. This was usually done on an exploratory basis to establish the need for continued monitoring in future patients. Three papers [34, 39, 40] analysed treatment effects on blood pressure, four papers [34, 36, 37, 40] analysed effects on lipid profiles and effects on insulin resistance. Treatment outcomes in this category are presented in Tables 7 and 8.

**Blood Pressure.** Three papers [34, 39, 40] analysed the effects of GD treatment on blood pressure–both systolic (SBP) and diastolic (DBP). One paper [34] analysed the effects of both GnRHa and CSH on blood pressure. Two papers [39, 40] looked at BP changes with GnRHa and testosterone use, and therefore only included NF.

For GnRHa-induced PS, two [34, 39] reported a significant increase in DBP (Klaver et al. [34] found only an effect in NM, with no effect in NF), whereas one (Stoffers et al. [40]) reported no change. There were no significant changes in SBP recorded in any of the papers during GnRHa treatment. There were no cases of hypertension with GnRHa reported in any of the included papers. From the papers that analysed the effects of CSH the results are similarly inconclusive. One paper [39] reported a decrease in DBP with testosterone to pre-

**Table 5. Age at treatment initiation–puberty suppression.**

| Location | Reference | Date[a] | N | NF or NM | Age at Treatment Initiation (years) | | | Mean duration (months)[b] | Age at last treatment (years) | |
|---|---|---|---|---|---|---|---|---|---|---|
| | | | | | Mean | SD | Range | | Mean (SD) | Range |
| Belgium | Tack et al. (2016) | 2010–15 | 38 | NF | 15.8 | - | - | 12.6 | - | - |
| | Tack et al. (2017) | 2008–16 | 27 | NM | 16.5 | - | - | 12.0 | - | - |
| | Tack et al. (2018) | 2011–17 | 21 | NM | 16.3 | 1.2 | - | 10.6 | - | - |
| | | as above | 44 | NF | 16.2 | 1.0 | - | 11.6 | - | - |
| Germany | Becker-Hebly et al. (2020) | 2013–17 | 11 | Both | 15.6 | 1.8 | 11.2–17.3 | - | 15.9 (1.84) | 11.7–17.6 |
| Israel | Perl et al. (2020) | 2013–18 | 15 | NF | 14.4 | 1.0 | - | - | 14.8 (1.0) | - |
| Netherlands | De Vries et al. (2011) | 2000–08 | 70 | Both | 14.7 | 1.9 | 11.3–18.6 | - | - | - |
| | Klaver et al. (2018) | 1998–15 | 71 | NM | 14.5 | 1.8 | - | 25.2[c] | - | - |
| | | as above | 121 | NF | 15.3 | 2.0 | - | 12.0[c] | - | - |
| | Klaver et al. (2020) | 1998–15 | 71 | NM | 14.6 | 1.8 | - | 25.2[c] | - | - |
| | | as above | 121 | NF | 15.2 | 2.0 | - | 12.0[c] | - | - |
| | Schagen et al. (2020) | 1998–2009 | 51 | NM | 14.1 | 1.7 | - | 24.0[c] | - | - |
| | | as above | 70 | NF | 14.5 | 2.0 | - | 21.6[c] | - | - |
| | Stoffers et al. (2019) | 2010–18 | 62 | NF | 16.5 | - | 11.8–18.0 | 8.0 | - | - |
| USA | Chen et al. (2016) | 2002–15 | 15 | Both | 15.5 | 2.1 | - | - | - | - |
| | Jensen et al. (2019) | 2016–18 | 6 | NM | 14.5 | - | 11.4–15.7 | 20.6[c d] | - | - |
| | | as above | 11 | NF | 13.9 | - | 12.9–15.6 | 29.3[c d] | - | - |
| | Kuper et al. (2020) | 2014–18 | 25 | Both | 13.7 | 1.5 | 9.8–14.9 | - | - | - |
| | Lee et al. (2020) | - | 33 | NM | 12.1 | 1.3 | 11.7–12.6 | - | - | - |
| | | - | 30 | NF | 11.0 | 1.4 | 10.5–11.5 | - | - | - |
| | Lopez et al. (2018) | 2004–16 | 40 | Both | - | - | 8.8–17.8[e] | - | - | - |
| | Nahata et al. (2017) | 2014–16 | 8 | Both | 15.3 | - | 12.8–17.3 | - | - | - |
| UK | Costa et al. (2015) | 2010–14 | 201 | Both | 16.5 | 1.3 | 13–17 | - | - | - |
| | Joseph et al. (2019) | 2011–16 | 70 | Both | 13.2 | 1.4 | 12–14 | - | - | - |
| | Russell et al. (2020) | - | 122 | Both | 13.6 | 0.1 | 9.9–15.9[f] | - | - | - |

- = not reported

Shaded = excluded from pooled mean age calculation due to likely, or stated, overlap in sample with other studies, or no SD reported

[a] = Date of data collection

[b] = duration of treatment at point of measures being taken, not necessarily total duration

[c] = refers explicitly to duration of GnRHa monotherapy

[d] = median reported

[e] = included only those subsamples under the age of 18

[f] = indicates age at which consent for PS taken–authors are clear this may not reflect the age of prescription / use of PS

treatment levels following a rise during GnRHa treatment. Stoffers et al. [40] reported a significant increase in SBP with testosterone use, however this followed an observed reduction over the preceding GnRHa treatment period so in fact represented a return to baseline. Klaver et al. (2020) [34] reported a significant increase in both DBP and SBP after addition of testosterone. Additionally, they reported a significant increase in DBP on the addition of oestrogen. It should be noted that Klaver et al. (2020) [34] measured the endpoint data in their cohort at the age of 22 years, and so is the only source of long-term data present. Overall, BP does not seem to be adversely affected by either GnRHa or CSH treatment. At no point in any of these papers did the BP measurements stray from the normal range.

**Lipid Profile.** The effects of GD treatment on lipid profile was analysed in four papers; one [34] looked at effects of GnRHa and CSH in a both NF and NM, one looked at the effects

**Table 6. Age at treatment initiation–cross sex hormones.**

| Location | Reference | Date[a] | N | NF or NM | Age at Treatment Initiation (years) | | | Mean duration (months)[b] | Age at last treatment (years) | |
|---|---|---|---|---|---|---|---|---|---|---|
| | | | | | Mean | SD | Range | | Mean (SD) | Range |
| Belgium | Tack et al. (2016) | 2010–15 | 43 | NF | 17·4 | - | - | 11.4 | - | - |
| | Tack et al. (2017) | 2008–16 | 27 | NM | 17·6 | - | - | 16.0 | - | - |
| Germany | Becker-Hebly et al. (2020) | 2013–17 | 32 | Both | 15.5 | 1.0 | 13.5–17.2 | - | 16.7 (1.2) | 14.5–19.8 |
| Israel | Perl et al. (2020) | 2013–18 | 9 | NF | 15·1 | 0.9 | - | - | 15.8 (0.9) | - |
| Netherlands | De Vries et al. (2011) | 2000–08 | 70 | Both | 16·6 | 1.9 | 13.9–19.2 | - | - | - |
| | Klaver et al. (2018) | 1998–15 | 71 | NM | 16·4 | 1.0 | - | 37.2[c] 33.6[d] | - | - |
| | | as above | 121 | NF | 16·9 | 0.9 | - | 28.8[c] 36.0[d] | - | - |
| | Klaver et al. (2020) | 1998–15 | 71 | NM | 16·4 | 1.0 | | 37.2[c] 27.6[d] | - | - |
| | | as above | 121 | NF | 16·9 | 0.9 | | 26.4[c] 34.8[d] | - | - |
| | Schagen et al. (2020) | 1998–2009 | 36 | NM | 16·2 | 1.2 | - | - | - | - |
| | | as above | 42 | NF | 16·9 | 1.1 | - | - | - | - |
| | Stoffers et al. (2019) | 2010–18 | 62 | NF | 17·2 | - | 14.9–18.4 | 12.0 | - | - |
| USA | Chen et al. (2016) | 2002–15 | 8 | Both | 16·2 | 1.4 | - | - | - | - |
| | Jensen et al. (2019) | 2016–18 | 16 | NM | 16·7 | - | 14.4–18.2 | 29.3 | - | - |
| | | as above | 50 | NF | 16·9 | - | 13.4–22.1 | 30.4[d] | - | - |
| | Kuper et al. (2020) | 2014–18 | 123 | Both | 16·2 | 1.2 | 13.2–18.6 | 10.9 | - | - |

Shaded = excluded from pooled mean age calculation due to likely, or stated, overlap in sample with other studies

a = Date of data collection

b = duration of treatment at point of measures being taken, not necessarily total duration

c = refers to duration of combined GnRHa and CSH treatment

d = refers to CSH monotherapy

of GnRHa and testosterone [40], one looked at the effects of lynestrenol (a progestin) and testosterone [36], and one looked at the effects of cyproterone acetate (an antiandrogen and progestin) and oestradiol [37].

Of the two papers that included data on the effects of GnRHa, one [34] found that GnRHa produced a significant increase in total cholesterol, low-density lipoprotein (LDL) and high-density lipoprotein (HDL). There was no significant change in triglycerides. The other [40] found that GnRHa treatment produced no significant effect on lipid profile. For CSH treatment, both papers found that testosterone produced a significant decrease in HDL. Stoffers et al. [40] found that other lipid parameters were unaffected, whereas Klaver et al. (2020) [34] noted a significant increase in total cholesterol, LDL and triglycerides. They also noted that oestrogen produced a significant increase in triglycerides. These changes were compared to trends amongst the non-GD population, and were found to be similar, with no significant differences between treated NF and non-treated cisgender men, and likewise for NM and non-treated cisgender women.

Tack et al. (2016) [36] found that lynestrenol produced a significant increase in LDL in the first 6 months of treatment, which then stabilised. It also produced a significant decrease in HDL. There were no significant changes to lipid profile with the addition of testosterone. Tack et al. (2017) [37] found that cyproterone acetate produced a significant decrease in total cholesterol, HDL and triglycerides. The addition of oestradiol produced no significant changes in lipid profile.

**Table 7. Treatment outcomes–blood pressure, biochemistry and haematology–puberty suppression.**

| Reference | Treatment | Sample (n) | Blood Pressure | Lipid Profile | Glucose metabolism | Liver Enzymes | Haemoglobin |
|---|---|---|---|---|---|---|---|
| Klaver et al. (2020) | GnRHa | 192 (71 NM) | Incr. SBP & DBP* | Incr. in Total*, HDL, LDL cholesterol. No change to Triglycerides | No change to Glucose, Insulin or HOMA-IR | - | - |
| Perl et al. (2020) | GnRHa | 15 (NF) | Incr. SBP & DBP* | - | - | - | - |
| Stoffers et al. (2019) | GnRHa | 62 (NF) | Decr. SBP, Incr. DBP | Incr. Total and LDL cholesterol. No change HDL, Triglycerides | No sig. changes to HbA1c | Incr. in ALP No significant change in ALT, AST, GGT | No change |
| Tack et al. (2016) | Lynestrenol | 39 (NF) | - | Decr. in HDL*, Incr. in LDL*, No change Total or Triglycerides | No sig. changes in HbA1c, glucose, insulin, or HOMA-IR | Incr. in ALT* and AST | Incr. at 6** and 12** months compared to T0 |
| Tack et al. (2017) | Cyproterone Acetate (CA) | 27 (NM) | | Decr. Total*, HDL*, Triglycerides. No change LDL. | No sig. changes in HbA1c, glucose, insulin, or HOMA-IR | No significant change in ALT and AST | Decr. At 6** and 12** months compared to T0 |

* / **- p<0.05 / p≤0.001; otherwise any change noted is non-significant

GnRHa = gonadotropin-releasing hormone analogues; NF = natal female; NM = natal male; decr. = decrease; incr. = increase; sig. = significant; ALP = alkaline phosphatase; ALT = alanine aminotransferase; AST = aspartate aminotransferase; DBP = diastolic blood pressure; GGT = gamma-glutamyl transferase; HbA1c = haemoglobin A1c; HOMA-IR = Homeostatic Model assessment of insulin resistance; SBP = systolic blood pressure; HDL = high-density lipoprotein; LDL = low-density lipoprotein.

**Table 8. Treatment outcomes–biochemical and haematological–cross sex hormones.**

| Reference | Treatment | Sample (n) | Blood Pressure | Lipid Profile | Insulin Resistance | Enzymes | Haemoglobin |
|---|---|---|---|---|---|---|---|
| Klaver et al. (2020) | Testosterone | 121 (NF) | Incr. SBP*, Incr. DBP* | Incr. in Total*, LDL, Triglycerides, Decr. in HDL* | No change in Glucose, Decr. in Insulin*, and HOMA-IR* | - | - |
| | Oestradiol | 71 (NM) | Decr. SBP, Incr. DBP* | Incr. in Triglycerides*, No change Total, HDL, LDL | No change to Glucose or HOMA-IR. Incr. in Insulin | - | - |
| Perl et al. (2020) | Testosterone | 9 (NF) | Decr. SBP, Decr. DBP | - | - | - | - |
| Stoffers et al. (2019) | Testosterone | 62 (NF) | Incr. SBP*, No change DBP | Decr. HDL*, Initial decr. Total*, subsequent incr., No change LDL, Triglycerides | No significant changes to HbA1c | Incr. in ALP* in first 6 months | Incr. at 6*, 12*, 24** months compared to T0 |
| Tack et al. (2016) | Lynrestrenol + Testosterone | 39 (NF) | - | No change Total, LDL, HDL, Triglycerides | No sig. changes in HbA1c, glucose, insulin or HOMA-IR | Incr. in ALT and AST | Incr. at 6** and 12** months compared to T0 |
| Tack et al. (2017) | Cyproterone Acetate + Oestradiol | 27 (NM) | - | No change Total, LDL, HDL, Incr. in Triglycerides | No sig. changes in HbA1c, glucose, insulin or HOMA-IR | Incr. in ALT. No significant change in AST. | No change |

* / **- p<0.05 / p<0.001; otherwise any change noted is non-significant

NF = natal female; NM = natal male; dec. = decrease; sig. = significant; ALP = alkaline phosphatase; ALT = alanine aminotransferase; AST = aspartate aminotransferase; DBP = diastolic blood pressure; GGT = gamma-glutamyl transferase; HbA1c = haemoglobin A1c; HOMA-IR = Homeostatic Model assessment of insulin resistance; SBP = systolic blood pressure; HDL = high-density lipoprotein; LDL = low-density lipoprotein.

**Glucose metabolism.**   Four papers measured parameters indicating glucose metabolism (GnRHa and CSH [34], testosterone [40], and progestins and CSH [36, 37]). They measured blood glucose levels, insulin levels and calculated the Homeostatic Model assessment of insulin resistance (HOMA-IR). The two Tack et al. [36, 37] papers also measured haemoglobin A1c (HbA1c) levels. Stoffers et al. [40] only measured HbA1c levels. These three papers noted no significant changes in any measure of insulin sensitivity with either PS (GnRHa or Progestin) or CSH. Klaver et al. (2020) [34] followed up their population into adulthood (22y) and found that there was a significant decrease in insulin and HOMA-IR after testosterone treatment. When compared to cisgender men, it was not significantly different. They did find that HOMA-IR of NM was higher than that of cisgender women. It must be noted that only two [34, 36] papers provided data, whereas the other two [37, 40] only mentioned the non-significance of their findings in the text.

**Haemoglobin.**   Three papers [36, 37, 40] measured the haemoglobin (Hb) of participants. The normal range is higher in men than in women. Tack et al. [36] (2016) analysed the effects of lynestrenol, finding a significant increase in mean Hb values. Two papers [36, 40] analysed the effects of testosterone and found increases in mean Hb, with Stoffers et al. [40] describing six individuals with haematocrit exceeding the upper male limit. These six were on an accelerated hormone program, and their values normalised by the end of follow-up without intervention or change of therapy. Tack et al. (2017) [37] analysed the effects of cyproterone acetate and oestradiol, finding a significant decrease in Hb with cyproterone acetate, and no further changes with oestradiol.

**Liver Enzymes.**   Four papers analysed changes in liver enzymes. Jensen et al. [48] simply noted whether there was any change but did not provide values nor which enzymes were measured. Two studies [36, 40] analysed the effects of testosterone and found differing increases in enzymes. Stoffers et al. [40] analysed alanine aminotransferase (ALT), aspartate aminotransferase (AST), gamma-glutamyl transferase (GGT) and alkaline phosphatase (ALP) and noted a significant increase only in ALP. Tack et al. (2016) [36] noted an increase in ALT and AST. These increases in ALT and AST remained within reference ranges for men. Stoffers et al. [40] had used GnRHa for PS, whereas Tack et al. (2016) [36] had used Lynestrenol. They did note that with Lynestrenol there was a significant increase in ALT, however this remained within reference ranges. Additionally, one patient's ALT went above the upper limit, yet this normalised with the addition of testosterone. Two studies [37, 48] analysed the effects of oestrogen. Jensen et al. [48] noted that two NMs had raised liver enzymes during oestrogen hormone therapy (although no values reported). Tack et al. (2017) [37], who reported the effects of cyproterone acetate and oestrogen, noted no significant increases in AST or ALT. They did note that five NMs had transiently increased levels, but none reach the threshold of 3-times the original value, which would indicate a need to stop treatment.

**Anthropometric, bone density, and physical changes.**   Height / weight / BMI

Seven studies included data on height and weight and / or Body Mass Index (BMI) [33, 35–37, 39, 40, 44] in relation to PS (Table 9). Findings were mixed, with no remarkable changes to anthropometry noted in any of the studies. Where weight or height increased it was generally in line with normal development for age and affirmed gender. An exception was reported in Joseph et al. [44] where NM had a higher BMI at baseline which increased more than that for NF. This was from a very small sample, however (n = 10).

Six papers reported data on anthropometry in relation to CSH [33, 34, 36, 37, 39, 40], with two of these being explicitly on the same sample at different time points (Klaver's studies [33, 34]) (see Table 10). More changes were noted for this stage of treatment, although generally in line with normal development for affirmed gender. Klaver (2020) noted a higher prevalence of

**Table 9. Treatment outcomes–anthropometric, bone density and physical changes–puberty suppression.**

| Reference | Treatment | Sample (n) | Height / Weight / BMI | Bone mineral density | Physical changes |
|---|---|---|---|---|---|
| Jensen et al. (2019) | GnRHa | 11 (NF) | - | - | Acne, mood changes, headache, hot flashes, injection site rash. |
| | | 6 (NM) | - | - | Breast tenderness |
| Joseph et al. (2019) | GnRHa | 31 (3y follow-up) | Gradual increase, greater BMI incr in NM, greater height incr in NF | No change BMD, decrease BMD z-scores* No change BMAD, decrease BMAD z-scores | - |
| | | 70 (2y follow-up) | - | No change BMD, decrease BMD z-scores* No change BMAD, decrease BMAD z-scores | - |
| Klaver et al. (2018) | GnRHa | 121 (NF) | incr TBF, decr LBM, decr WHR | - | - |
| | | 71 (NM) | incr TBF, decr LBM, decr WHR–larger change than NF | - | - |
| Perl et al. (2020) | GnRHa | 15 (NF) | No significant change in weight status (BMI-SDS) | - | - |
| Schagen et al. (2020) | GnRHa | 15 (prolonged GnRHa - 3y) | - | No change in BMD, decrease in BMD z-scores | - |
| | | 51 (NM) | - | No change BMD, decrease BMD z-scores* No change BMAD, decrease BMAD z-scores* | - |
| | | 70 (NF) | - | No change BMD in early pubertal group, dec. BMD z-scores* Decrease BMD* in late pubertal group, dec. BMD z-scores* Decrease BMAD*, decrease BMAD z-scores* | - |
| Stoffers et al. (2019) | GnRHa | 62 (NF) | No significant change in BMI with ref to male and female SDS | Decrease in BMD*, BMD z-scores* | Increased hair growth and voice deepening within 3 months of testosterone treatment. |
| Tack et al. (2016) | Lynestrenol | 38 (NF) | Weight, BMI incr in 1st 6m, returned to baseline after 12m | - | Headache, hot flushes, acne, metrorrhagia (but dropped after 6m), fatigue |
| Tack et al. (2017) | Cyproterone Acetate | 27 (NM) | No clinically important changes in body weight and BMI | - | Decr. facial and non-facial hair growth, breast development, fatigue |
| Tack et al. (2018) | Lynestrenol | 44 (NF) | lean mass and grip strength sig incr | Increase BMD, no change in z-scores | - |
| | Cyproterone Acetate | 21 (NM) | loss of lean mass, gain of fat mass, decr grip strength Z scores. | Decrease BMD, and BMD z-scores* | - |

*- denotes significant change (p<0.05), otherwise any change noted is non-significant

GnRHa = gonadotropin-releasing hormone analogues; NF = natal female; NM = natal male; decr. = decrease; incr. = increase; sig. = significant; BMI = Body Mass Index; TBF = Total body fat; LBM = lean body mass; WHR = waist to hip ratio; BMD = bone mineral density; BMAD = bone mineral apparent density; SDS = standard deviation score.

obesity in their follow-up at 22 years old in both NM and NF participants. Tack et al. (2016) [36] also noted significant weight gain in NF (but not in NM in their 2017 paper [37]).

Other measures such as waist to hip ratio, lean body mass and body fat percentage were employed by some papers, but no remarkable findings were noted.

**Bone density.** Five papers contained data relating to bone health of which one (Lee et al. [41]) focussed on a pre-treatment population (n = 63) and so was not included in this analysis. The four remaining papers all measured areal bone mineral density (BMD or aBMD; g/cm$^2$) with three also measuring bone mineral apparent density (BMAD; g/cm$^3$)—a size adjusted value incorporating body size measurements—using dual energy X-ray absorptiometry (DXA)

**Table 10. Treatment outcomes–anthropometric, bone density and physical changes–cross sex hormones.**

| Reference | Treatment | Sample (n) | Height / Weight / BMI | BMD and/or BMAD | Physical features |
|---|---|---|---|---|---|
| Jensen et al. (2019) | Testosterone | 50 (NF) | - | - | Acne, mood changes, headache, increased appetite, fatigue, hair loss, hot flashes, spotting. |
| | Oestradiol | 16 (NM) | - | - | Breast tenderness |
| Klaver et al. (2018) | Testosterone | 121 (NF) | Changes according to affirmed sex (decr TBF*, incr LBM*, incr WHR*) TBF, LBM, WHR between ref values for ciswomen & cismen | - | - |
| | Oestradiol | 71 (NM) | Changes according to affirmed sex (incr TBF*, decr LBM*, decr WHR*) TBF, LBM, WHR most similar to ciswomen | - | - |
| Klaver et al. (2020) | Testosterone | 121 (NF) | increase in obesity prevalence (+1.6%) comparable with cismen (+1.2%). At 22 years of age, the prevalence of obesity was higher (6.6%) than in cismen (3.0%) or ciswomen (2.2%). | - | - |
| | Oestradiol | 71 (NM) | increase of obesity prevalence (+8.5%) was more remarkable compared with ciswomen (+0.7%). At 22 years of age, the prevalence of obesity was higher (9.9%) than in cismen (3.0%) or ciswomen (2.2%). | - | - |
| Perl et al. (2020) | Testosterone | 9 (NF) | No significant change in weight status (BMI-SDS) | - | |
| Schagen et al. (2020) | Testosterone | 42 (NF) | - | Increase BMD*, increase BMD z-scores* Increase BMAD*, increase BMAD z-scores* | - |
| | Oestradiol | 36 (NM) | - | Increase BMD*, increase BMD z-scores* Increase BMAD*, increase BMAD z-scores* | - |
| Stoffers et al. (2019) | Testosterone | 62 (NF) | BMI sig. incr. in first 6m of treatment, but no different to male and female SDS | Increase in BMD*, BMD Z-scores* Not significantly different from baseline. Z-scores still below 0 | - |
| Tack et al. (2016) | Lynrestrenol + Testosterone | 39 (NF) | significant and continuous weight gain after 6m & 12m | - | Acne, metrorrhagia, fatigue |
| Tack et al. (2017) | Cyproterone Acetate + Oestradiol | 27 (NM) | No clinically important changes in body weight and BMI | - | Breast development, resolution of fatigue, breast tenderness, emotionality, hunger |

*- denotes significant change (p<0.05), otherwise any change noted is non-significant

NF = natal female; NM = natal male; decr. = decrease; incr. = increase; sig. = significant; BMI = Body Mass Index; TBF = Total body fat; LBM = lean body mass; WHR = waist to hip ratio; BMD = bone mineral density; BMAD = bone mineral apparent density.

scanning. These papers also all compared obtained Z-scores with reference ranges of age-matched peers of the same birth sex as the GD adolescents being studied (i.e., compared Z-scores of NF with cisgender females).

BMD in relation to GnRHa treatment was examined in three papers. Two papers reported that the absolute values for BMD did not change significantly [44, 51] with GnRHa usage, whereas Stoffers et al. [40] reported that BMD significantly decreased compared to pre-treatment values. The absolute values of BMAD significantly decreased in one paper [51] but did not change significantly in the other [44]. GnRHa treatment resulted in significant decreases in both BMD and BMAD Z-scores across all three papers. Additionally, Schagen et al. [51]

reported on a small cohort (n = 15) of four NMs (mean age 12.6) and 11 NFs (mean age 12.7) who were on prolonged (3 years) GnRHa treatment and found that absolute BMD values at the lumbar spine and hip remained stable, but z-scores did decline.

CSH effects on bone health were investigated in two papers [40, 51]. Both papers identified a significant increase in BMD absolute values after hormonal treatment, with Schagen et al. [51] also noting a significant increase in BMAD absolute values. Both noted significant increases in BMD Z-scores, with the Z-scores returning close to 0. However, both described those values in their NM groups remained significantly below 0.

Tack et al. (2018) [35] analysed at the effects of lynestrenol and cyproterone acetate rather than GnRHa. In the lynestrenol group absolute aBMD values remained stable. There was a significant increase in BMD z-scores in the total hip area, and z-scores remained stable in the femoral neck and lumbar spine. In the cyproterone acetate group it was found that aBMD remained stable at the femoral neck and lumbar spine but decreased significantly in the hip. There was also a significant decrease in Z-scores of total hip, femoral neck and lumbar spine.

**Physical changes.** Only four papers [36, 37, 40, 48] reported other physical changes associated with treatment. The most common mentioned effects were amenorrhea, acne, weight gain, hair loss, headaches, fatigue, hot flushes, breast tenderness, mood swings and changes in appetite. Tack et al. (2016) indicated that acne prevalence increased during the first six months of combined Lynestrenol and testosterone treatment, and that metrorrhagia declined in the following 6 months of Lynestrenol treatment.

**Mental health outcomes.** Five included papers covered mental health outcomes [38, 43, 45, 49, 50], which are summarised in Table 11. Three of the studies used the Children's Global Assessment Scale (CGAS [52]), a clinician rating to assess global functioning where high scores (>80) indicate good global functioning. In general, adolescents with GD showed some problems, without severe impairment, on this scale: baseline scores ranged between 55–74. Becker-Hebly et al. (2020) indicated that functioning improved to 'good' (scores between 81–85) after PS monotherapy, PS+CSH combined therapy, and surgical intervention. However, these scores were not compared to the German norm. Costa et al. (2015) reported that adolescents had some problems with global functioning at baseline which significantly improved after 12 months of PS, better than those receiving psychological support alone, and similar to a comparison group. de Vries et al. (2011) reported a significant improvement in global functioning after an average of just under 2 years' PS treatment.

Most measures of mental health status were self- or parent-report. Becker-Hebly et al. (2020) used the Youth / Adult Self Report (YSR / ASR [53, 54]) to measure emotional and behavioural problems and observed that scores at baseline were lower compared to the German norm i.e., patients had a high prevalence of problems, and scores barely improved after any level of treatment. In their paper, de Vries et al. (2011) reported that YSR scores decreased significantly, and the percentage of adolescents that scored within the clinical range on the internalising scale decreased significantly following PS. The Body Image Scale (BIS [55]) was administrated in 2 studies: de Vries et al. (2011) noted a higher dissatisfaction in NF than in NM for secondary sex characteristics at follow-up, although no significant changes were noted in BIS over time following PS; Kuper et al. (2020) reported that BIS scores were significantly lower at follow-up in both PS and CSH groups (no data were reported for surgical intervention). Becker-Hebly et al. (2020) utilised the Kidscreen-27 [56] to asess mental dimensions of quality of life. Baseline scores in all groups were below the German norm, with the PS group scoring within the norm at follow-up for both mental and physical quality of life. The other two groups (PS+CSH and surgery) also showed improved quality of life at follow-up, but only the physical health dimensions were within German norms (mental health scores remained lower).

**Table 11. Treatment outcomes–mental health–all treatment types.**

| Reference | Treatment | Sample (n) | Mental health outcomes |
|---|---|---|---|
| Becker-Hebly et al. (2020) | Psychosocial only | 21 (3 NM) | All YSR/ASR scores signif higher than norm @ BL & FU<br>QoL signif below mean @ BL, below norm @ FU<br>CGAS improved BL to FU (no norm ref) |
| | GnRHa only | 11 (3 NM) | All YSR/ASR scores signif higher than norm @ BL, only Total & Internalising subscales higher than norm @ FU<br>QoL signif below mean @ BL, within norm @ FU<br>CGAS improved BL to FU (no norm ref) |
| | CSH + GnRHa | 32 (4 NM) | YSR/ASR Total & Internalising scores signif higher than norm @ BL, only Total scores higher than norm at FU<br>QoL signif below mean @ BL, physical QoL within norm @ FU<br>CGAS improved BL to FU (no norm ref) |
| | Surgery | 11 (1 NM) | YSR/ASR Total & Internalising scores signif higher than norm @ BL, only Total scores higher than norm at FU<br>QoL signif below mean @ BL, physical QoL within norm @ FU<br>CGAS improved BL to FU (no norm ref) |
| Costa et al. (2015) | Psychological support only (Delayed eligible GnRHa) | 100 | CGAS scores signif higher @ 6m; no further signif improvement. Remained lower than comparison grp after 18m |
| | GnRHa (Immediately eligible) | 101 | CGAS scores–no improvement @6m; signif improved @12m and similar to comparison grp |
| de Vries et al. (2011) | GnRHa | 70 (37 NF; 33 NM) | CBCL: Signif decrease in T-score 44% scoring in clinical range but decreased to 22% T0 –T1<br>YSR: Signif decrease in T-score 29.6% scoring in clinical range on internalising scale but decreased to 11.1% T0 –T1<br>CGAS: Signif improvement T0 –T1<br>BDI-II: Signif decrease T0 –T1<br>No significant changes in STAI, UGDS or BIS T0 –T1<br>More body dissatisfaction in NF than in NM in UGDS and BIS–for secondary sex characteristics |
| Kuper et al. (2020) | GnRHa | 25 | BIS Body image: decreased at FU<br>QIDS Depression: self-report decreased at FU (within 'mild' range); clinical report: no significant changes (within 'mild' range).<br>SCARED Anxiety: decreased at FU |
| | CSH | 123 | BIS Body image: decreased score at FU<br>QIDS Depression: self-report decreased at FU (within 'mild' range); clinical report: no significant changes (within 'mild' range).<br>SCARED Anxiety: decreased at FU |
| | Surgery | 15 | No data presented |
| Russell et al. (2020) | GnRHa | 95 (57 NF; 38 NM) | SRS2: No significant change in scores from BL to FU. |

GnRHa = gonadotropin-releasing hormone analogues; CSH = cross sex hormones.

NF = natal female; NM = natal male; decr. = decrease; incr. = increase; sig. = significant; BL = baseline; FU = follow-up.

ASR = Adult Self-Report; BID-II = Beck Depression Inventory; BIS = Body Image Scale; CBCL = Child Behavior Checklist; CGAS = Children's Global Assessment Scale; QIDS = Quick Inventory of Depressive Symptoms; QoL = Quality of life; SCARED = Screen for Childhood Anxiety Related Emotional Disorders; SRS2 = Social Responsiveness Scale; STAI = State Trait Anxiety Inventory; UGDS = Utrecht Gender Dysphoria Scale; YSR = Youth Self-Report.

de Vries et al. (2011) measured general behaviour disruption with the Child Behaviour Checklist (CBCL), showing a significant decrease in T-scores following PS, as well as a decrease in the percentage of adolescents that scored within the clinical range. These authors also described a significant decrease in depressive symptoms, when assessed with the Beck Depression Inventory (BDI-II [57]). Kuper et al. (2020) observed that both self-reported and clinical-reported depressive symptoms measured by the Quick Inventory of Depressive Symptoms (QIDS [58]) were within 'mild' range (0–27) at follow-up for both PS and CSH groups.

Anxiety symptoms, measured by SCARED [59], also decreased at follow-up. Finally, Russell et al. (2020) found no change in autism symptoms using the Social Responsiveness Scale (SRS-2 [60]) over time undergoing PS.

Gender dysphoria symptoms were measured at follow-up in 1 sample: de Vries et al. (2011) reported no change from baseline to follow-up undergoing PS using the Utrecht Gender Dysphoria Scale (UGDS) [38].

**What are the long-term outcomes for all (treated or otherwise) in this population?.**
None of the included papers featured long-term follow-up. One paper, Klaver et al. (2020) [34], included follow-up at 22 years old, with the only remarkable finding being an increase in BMI and obesity beyond the cisgendered population norm.

## Quality assessment

The CCAT quality ratings ranged from 71% to 95%, with a mean of 82%. All papers achieved an overall rating of 4 (good, n = 8) or 5 (very good, n = 11), with strengths and weaknesses within certain discrete categories; Papers tended to score higher in Introduction, Preliminaries and Data Collection. One area that many papers scored lower on was the issue of consent—due to many being retrospective in nature some centres waived the requirements of the researchers to seek consent, and some simply did not mention a consent process at all. See Table 12 for full data.

**Table 12. Quality ratings using Crowe Critical Appraisal Tool (CCAT).**

| | | | Average CCAT rating | | | | | | | | | | |
|---|---|---|---|---|---|---|---|---|---|---|---|---|---|
| ID | Country | Reference | 1 | 2 | 3 | 4 | 5 | 6 | 7 | 8 | | | |
| | | | Preliminaries | Introduction | Design | Sampling | Data collection | Ethical matters | Results | Discussion | Total | % | Overall level |
| 1 | Belgium | Tack et al. (2016) | 4 | 4·5 | 3·5 | 4 | 4·5 | 4·5 | 3·5 | 3 | 31·5 | 79 | 4 |
| 2 | Belgium | Tack et al. (2017) | 4·5 | 4·5 | 4 | 4 | 4·5 | 5 | 4 | 3·5 | 34 | 85 | 5 |
| 3 | Belgium | Tack, et al. (2018) | 4·5 | 4 | 3·5 | 3·5 | 4·5 | 4·5 | 4 | 4 | 32·5 | 81 | 5 |
| 4 | Germany | Becker-Hebly et al. (2020) | 4·5 | 4·5 | 4 | 4·5 | 3·5 | 4 | 4 | 4·5 | 33·5 | 84 | 5 |
| 5 | Israel | Perl et al. (2020) | 4·5 | 4 | 4 | 3·5 | 4·5 | 3·5 | 3·5 | 4 | 31·5 | 79 | 4 |
| 6 | N/lands | de Vries, et al. (2011) | 5 | 4 | 2·5 | 3·5 | 4·5 | 4 | 4 | 3 | 30·5 | 76 | 4 |
| 7 | N/lands | Klaver, et al. (2018) | 4·5 | 4·5 | 5 | 5 | 5 | 5 | 4·5 | 4·5 | 38 | 95 | 5 |
| 8 | N/lands | Klaver, et al. (2020) | 4 | 4·5 | 4·5 | 4 | 4 | 4·5 | 4 | 4·5 | 34 | 85 | 5 |
| 9 | N/lands | Schagen, et al. (2020) | 4·5 | 5 | 4 | 4 | 4·5 | 4 | 3·5 | 4 | 33·5 | 84 | 5 |
| 10 | N/lands | Stoffers et al. (2019) | 4·5 | 4·5 | 3·5 | 4 | 5 | 4·5 | 4 | 5 | 35 | 88 | 5 |
| 11 | UK | Costa, et al. (2015) | 4·5 | 4·5 | 4 | 4·5 | 4 | 3·5 | 4 | 4·5 | 33·5 | 84 | 5 |
| 12 | UK | Joseph, et al. (2019) | 4 | 4 | 3·5 | 3·5 | 4 | 4 | 3·5 | 3 | 29·5 | 74 | 4 |
| 13 | UK | Russell, et al. (2020) | 5 | 5 | 3·5 | 3·5 | 4 | 2·5 | 3·5 | 3·5 | 30·5 | 76 | 4 |
| 14 | USA | Chen, et al. (2016) | 4 | 3·5 | 4 | 4 | 4 | 2·5 | 3·5 | 3 | 28·5 | 71 | 4 |
| 15 | USA | Jensen et al. (2019) | 4·5 | 4·5 | 3·5 | 4 | 4 | 4 | 4 | 4 | 32·5 | 81 | 5 |
| 16 | USA | Kuper, et al. (2020) | 5 | 5 | 4 | 4·5 | 5 | 3 | 4 | 3·5 | 34 | 85 | 5 |
| 17 | USA | Lee et al. (2020) | 5 | 4·5 | 4 | 4·5 | 4·5 | 2·5 | 4 | 5 | 34 | 85 | 5 |
| 18 | USA | Lopez, et al. (2018) | 4 | 4 | 4·5 | 4·5 | 4 | 4 | 3·5 | 3 | 31·5 | 79 | 4 |
| 19 | USA | Nahata, et al. (2017) | 4 | 4 | 4 | 4 | 4 | 4 | 4 | 3 | 31 | 78 | 4 |

## Discussion

This systematic review synthesises research evidence regarding the treatment type, age at treatment, and outcomes for adolescents presenting for assessment for gender dysphoria (GD). We identified 19 papers showing that most centres publishing data provided treatment according to WPATH guidelines relevant at the time (v7) [7]. Young people started PS, usually GnRHa, at a mean age of 14.5, and CSH at a mean age of 16.2 years, although there were very wide ranges around these central points, with children as young as 8 years old starting PS and 13 years old starting CSH. Surgical intervention was uncommon in this sample: 25 participants underwent mastectomy and one vaginoplasty (from 2 and 1 paper respectively), with the lower age range at 15 years old.

Most of the included papers covered a range of exploratory monitoring measures due to this field of study being novel and following a desire to ensure the safety of endocrine interventions with adolescents at a sensitive time in development. These measures mostly generated unremarkable findings: although some changes were observed through PS monotherapy, these tended to resolve once CSH was introduced, and physical development continued according to affirmed gender. Notable findings were in relation to BMD and obesity: BMD appears to decrease with PS but recover with CSH, but findings are heterogeneous; one paper including longer-term follow-up (to age 22 years) showed increased obesity prevalence in both NM and NF. Mental health was measured in a small number of papers, with indications of improvement over time in treatment. Where GD symptoms were measured there was no improvement at follow-up, but these data were from samples undergoing PS monotherapy, and so GD symptoms would reasonably not be expected to improve significantly at this stage (i.e., prior to developing secondary characteristics of identified gender).

A theme common to all three papers in this review series is the clear need for prospective research on large samples from broader populations and with long term follow-up in order to fully understand the implication of intervention during adolescence. We originally set out to study the phenomenon of adolescent- or rapid-onset GD (AOGD) and found an absence of literature, leading to our broader search strategy. There continues debate as to whether AOGD is a genuine phenomenon: Bauer et al. (2022) [61] provided data to suggest it is not, but faced strong rebuttal from both Littman (2022) [62] and Sinai (2022) [63] in terms of the way that AOGD has been defined and clinician experience. It is clear that we simply do not know enough about the observed phenomenon referred to as AOGD, nor do we fully understand the huge increase in numbers of adolescents (and especially NF) presenting for GD intervention in recent years, nor the comorbidities and long-term outcomes.

### Strengths and limitations

This review has strength in the broad search strategy and thorough hand screening process applied. There is also strength in this being part of a three-part comprehensive series. However, the limitation of this approach is that time has passed since the initial searches were conducted and new literature has been published which may change the final conclusions of this paper. We have chosen to curtail this final paper to the same end date as the preceding two in the series in the interests of consistency–these three papers should cover the same time period to be considered part of the same overarching review. We conducted a quick search according to our original strategy: this returned 2208 new records without de-duplication. Based on the screening process for the present review, we could expect this to yield about 10 further papers for inclusion. However, as pointed out in the interim report for the Cass review [24, 25], good quality evidence is most definitely still lacking. The Cass review [64] is now in a position to conduct more detailed systematic reviews based on a mandate from policy makers, which is a

huge step forward in developing this field of evidence and giving it the prominence it deserves. Although UK-based, the quality of this review will have implications for the field internationally. We do not think it would be fruitful to update our review in light of this developing work, although expect our paper series to provide a useful overview whilst the Cass review is ongoing.

The broad initial search criteria led to the need for some narrowing of criteria following initial screening (but prior to full-text screening). The addition of parameters regarding type of publication, upper age of participants, and the clinical verification of GD naturally narrowed the pool of papers and therefore may have meant papers with important findings have been excluded (for example, if a paper included an upper age limit of 21 even though the majority were younger than 18). We endeavoured to record all papers that only narrowly missed inclusion on the age criterion (Table 4), but literature that was excluded on the basis of type (i.e., conference proceedings and grey literature) were not included at this stage and so the potential contribution of this body of work cannot be quantified or assessed. We opted to use a quality assessment tool for studies of diverse designs (CCAT). This allowed all papers to be rated using the same system, but also involved reviewers having to make subjective ratings rather than apply a strictly quantifiable checklist. This may have led to issues with quality, such as over-statement of the significance of findings, not being sufficiently prominent.

Although we were able to include 19 papers from a range of countries in this review, just under a half (8) arose from two well-established treatment centres: those in Amsterdam and London. The Amsterdam team has led the way in developing assessment and treatment protocols for GD and provides a wealth of data over a long period (since 1996 within the included papers), and the London GIDS has, until very recently, been a hub for the whole of the United Kingdom now dealing with hundreds of referrals per year. This presents the advantage of being able to observe the adolescent GD population over a long period of time, assessed using the same or similar tools, and within a relatively stable social context. It is not clear, however, what proportion of young people experiencing GD have access to these national specialist centres and how many may be accessing private facilities or self-medicating with hormones obtained via other routes: we do not know how representative these samples are. Another disadvantage is that most of the papers included in this review are likely to include data from the same samples of participants, also limiting generalisability. The overlap between samples was rarely overtly stated, and there is a risk that readers may add greater weight to collective findings than is warranted. There is a clear lack of research on GD in low and middle income countries in the scientific literature [65], so the impact of different service contexts in countries such as India and Thailand cannot be properly considered [66].

## Conclusion

There is a lack of evidence on treatment for GD in adolescence. Although there is a growing body of literature providing data, there are limitations to the scope and quality, and prospective studies with long-term follow-up from a range of centres internationally is required. This review series has highlighted a lack of quality evidence in relation to adolescent GD in general: epidemiology, comorbidity, and treatment impact is difficult to robustly assess. Without an improvement in the scientific field, clinicians, parents, and young people are left ill-equipped to make safe and appropriate decisions.

## Acknowledgments

Special thanks to Ingrid Vinsa, Research Nurse and Administrator at the Gillberg Neuropsychiatry Centre, for her invaluable assistance in obtaining full text papers and assistance to CG in supervision of this piece of work.

## Author Contributions

**Conceptualization:** Lucy Thompson, Christopher Gillberg.

**Data curation:** Lucy Thompson, Darko Sarovic, Philip Wilson, Louis Irwin, Dana Visnitchi, Angela Sämfjord, Christopher Gillberg.

**Formal analysis:** Lucy Thompson, Louis Irwin, Dana Visnitchi.

**Investigation:** Lucy Thompson, Darko Sarovic, Philip Wilson, Angela Sämfjord.

**Methodology:** Lucy Thompson, Christopher Gillberg.

**Project administration:** Lucy Thompson, Dana Visnitchi.

**Supervision:** Philip Wilson, Christopher Gillberg.

**Validation:** Lucy Thompson, Darko Sarovic, Philip Wilson, Louis Irwin, Dana Visnitchi, Angela Sämfjord, Christopher Gillberg.

**Visualization:** Lucy Thompson.

**Writing – original draft:** Lucy Thompson, Philip Wilson, Louis Irwin, Dana Visnitchi, Angela Sämfjord.

**Writing – review & editing:** Lucy Thompson, Darko Sarovic, Philip Wilson, Louis Irwin, Dana Visnitchi, Angela Sämfjord, Christopher Gillberg.

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
