## [Decision Letter · Decision Letter 0]

26 Apr 2023

PGPH-D-22-01620

A PRISMA systematic review of adolescent gender dysphoria literature: 3) treatment

Dear Dr. Thompson,

Thank you for submitting your manuscript to PLOS Global Public Health. After careful consideration, we feel that it has merit but does not fully meet PLOS Global Public Health’s publication criteria as it currently stands. Therefore, we invite you to submit a revised version of the manuscript that addresses the points raised during the review process.

We look forward to receiving your revised manuscript.

Kind regards,

Vanessa Carels

Staff Editor

Journal Requirements:

2. Please upload a copy of Figure 2 which you refer to in your text on page 9. Or, if the figure is no longer to be included as part of the submission please remove all reference to it within the text.

Additional Editor Comments (if provided):

Reviewers' comments:

Reviewer's Responses to Questions

**Comments to the Author**

1. Does this manuscript meet PLOS Global Public Health’s publication criteria? Is the manuscript technically sound, and do the data support the conclusions? The manuscript must describe methodologically and ethically rigorous research with conclusions that are appropriately drawn based on the data presented.

Reviewer #1: Yes

Reviewer #2: Yes

2. Has the statistical analysis been performed appropriately and rigorously?

Reviewer #1: N/A

Reviewer #2: Yes

3. Have the authors made all data underlying the findings in their manuscript fully available (please refer to the Data Availability Statement at the start of the manuscript PDF file)?

Reviewer #1: Yes

Reviewer #2: Yes

4. Is the manuscript presented in an intelligible fashion and written in standard English?

Reviewer #1: Yes

Reviewer #2: Yes

5. Review Comments to the Author

Reviewer #1: 1&2. the manuscript has gone through rigorous exercise of systematic review though no statistics was applied to this review. Authors have provided an explanation for not including it. They have used a qualitative approach for this review.

3. Yes the authors have made the document available

4. There are grammatical and writing issues in this paper that I have mentioned in my review. Overall, standard of writing is acceptable but need improvement.

Reviewer #2: Really enjoyed reading this paper; it is a well balanced and well written review; it is a bit unfortunte that from the included data to draw definitive conclusions regarding the safety of treatment; I am interested in knowing the difference is GD rate between DSD population and non DSD population; is it possible to include that in the results/ discussion?

6. PLOS authors have the option to publish the peer review history of their article (what does this mean?). If published, this will include your full peer review and any attached files.

**Do you want your identity to be public for this peer review?** For information about this choice, including consent withdrawal, please see our Privacy Policy.

Reviewer #1: No

Reviewer #2: No

---

## [Decision Letter · Decision Letter 1]

30 Jun 2023

A PRISMA systematic review of adolescent gender dysphoria literature: 3) treatment

PGPH-D-22-01620R1

Dear Dr Thompson,

We are pleased to inform you that your manuscript 'A PRISMA systematic review of adolescent gender dysphoria literature: 3) treatment' has been provisionally accepted for publication in PLOS Global Public Health.

Best regards,

Julia Robinson

Executive Editor

Reviewer Comments (if any, and for reference):

Reviewer's Responses to Questions

**Comments to the Author**

1. If the authors have adequately addressed your comments raised in a previous round of review and you feel that this manuscript is now acceptable for publication, you may indicate that here to bypass the “Comments to the Author” section, enter your conflict of interest statement in the “Confidential to Editor” section, and submit your "Accept" recommendation.

Reviewer #1: All comments have been addressed

Reviewer #2: All comments have been addressed

2. Does this manuscript meet PLOS Global Public Health’s publication criteria? Is the manuscript technically sound, and do the data support the conclusions? The manuscript must describe methodologically and ethically rigorous research with conclusions that are appropriately drawn based on the data presented.

Reviewer #1: Yes

Reviewer #2: Yes

3. Has the statistical analysis been performed appropriately and rigorously?

Reviewer #1: N/A

Reviewer #2: N/A

4. Have the authors made all data underlying the findings in their manuscript fully available (please refer to the Data Availability Statement at the start of the manuscript PDF file)?

Reviewer #1: Yes

Reviewer #2: Yes

5. Is the manuscript presented in an intelligible fashion and written in standard English?

Reviewer #1: Yes

Reviewer #2: Yes

6. Review Comments to the Author

Reviewer #1: 1. All comments have been addressed pertaining to citations, clarity in sentences.

Reviewer #2: (No Response)

7. PLOS authors have the option to publish the peer review history of their article (what does this mean?). If published, this will include your full peer review and any attached files.

**Do you want your identity to be public for this peer review?** For information about this choice, including consent withdrawal, please see our Privacy Policy.

Reviewer #1: No

Reviewer #2: **Yes: **Ramesh Babu
